# A human endothelial cell-based recycling assay for screening of FcRn targeted molecules

Algirdas Grevys[1,2], Jeannette Nilsen[2,3], Kine M.K. Sand[1,2], Muluneh B. Daba[1,2], Inger Øynebråten[4], Malin Bern[1,2], Martin B. McAdam[1,2], Stian Foss[1,2], Tilman Schlothauer[5], Terje E. Michaelsen[6,7], Gregory J. Christianson[8], Derry C. Roopenian[8], Richard S. Blumberg[9], Inger Sandlie[1,2] & Jan Terje Andersen[1,2,10]

Albumin and IgG have remarkably long serum half-lives due to pH-dependent FcRn-mediated cellular recycling that rescues both ligands from intracellular degradation. Furthermore, increase in half-lives of IgG and albumin-based therapeutics has the potential to improve their efficacies, but there is a great need for robust methods for screening of relative FcRn-dependent recycling ability. Here, we report on a novel human endothelial cell-based recycling assay (HERA) that can be used for such pre-clinical screening. In HERA, rescue from degradation depends on FcRn, and engineered ligands are recycled in a manner that correlates with their half-lives in human FcRn transgenic mice. Thus, HERA is a novel cellular assay that can be used to predict how FcRn-binding proteins are rescued from intracellular degradation.

[1] Centre for Immune Regulation (CIR) and Department of Biosciences, University of Oslo, N-0316 Oslo, Norway. [2] CIR and Department of Immunology, Rikshospitalet, Oslo University Hospital and University of Oslo, PO Box 4950, N-0424 Oslo, Norway. [3] Institute of Clinical Medicine, University of Oslo, N-0450 Oslo, Norway. [4] Department of Pathology, Rikshospitalet, Oslo University Hospital and University of Oslo, N-0424 Oslo, Norway. [5] Biochemical and Analytical Research, Large Molecule Research, Roche Pharma Research and Early Development (pRED), Roche Innovation Center, DE-82377 Munich, Germany. [6] School of Pharmacy, University of Oslo, N-0371 Oslo, Norway. [7] Norwegian Institute of Public Health, Infection Immunology, N-0403 Oslo, Norway. [8] Jackson Laboratory, Bar Harbor, ME 04609, USA. [9] Division of Gastroenterology, Department of Medicine, Brigham & Women's Hospital, Harvard Medical School, 75 Francis St, Boston, MA 02115, USA. [10] Department of Pharmacology, Institute of Clinical Medicine, University of Oslo and Oslo University Hospital, N-0424 Oslo, Norway. Jeannette Nilsen, Kine M. K. Sand and Muluneh B. Daba contributed equally to this work. Correspondence and requests for materials should be addressed to J.T.A. (email: j.t.andersen@medisin.uio.no)

Since the development of the hybridoma technology around 40 years ago[1], monoclonal antibodies have become important therapeutics, particularly for the treatment of cancer and autoimmune diseases[2,3]. Antibody engineering initially focused on humanization as well as improvement of antigen affinity[4]. There has been more recent interest in fine-tuning of the Fc region[2,4]. The major antibody class found in the blood, IgG, has a remarkable persistence, with a serum half-life of 20–23 days, compared with only hours or a few days for other circulating proteins[5]. The only exception is albumin, which has a similar long half-life, and is utilized as carrier for therapeutics[6,7]. In both cases, the long half-life is due to their molecular size above the renal clearance threshold and their interaction with a membrane-bound receptor named the neonatal Fc receptor (FcRn).

FcRn is an MHC class I-related molecule that consists of a transmembrane heavy chain (HC) that non-covalently associates with β2-microglobulin[8–10]. The receptor binds both ligands non-competitively in a strictly acidic pH-dependent manner, with negligible binding and release at neutral pH[11–14]. While FcRn binds the CH2–CH3 elbow region of IgG, both domain I and III of albumin are required for optimal binding to the receptor[13,15,16].

Advanced imaging studies have demonstrated that FcRn is predominantly located within acidified endosomes, where the low pH allows binding of IgG taken up by fluid-phase pinocytosis[17]. FcRn then recycles its IgG to the cell surface for release into the circulation upon exposure to the physiological pH of the blood[18–20]. Proteins that do not bind the receptor are directed to lysosomal degradation. As albumin binds FcRn in a similar pH-dependent manner[12,14,21,22], recent data support that it follows the same recycling pathway[23]. Further, FcRn expressed by endothelial cells regulate both ligands, hematopoietic cells determine IgG homoeostasis while hepatocytes regulate albumin but not IgG[24–27].

Several studies have demonstrated the shortcomings of standard laboratory mice as pre-clinical models for evaluation of human IgG (hIgG) and human serum albumin (HSA) pharmacokinetics, due to considerable differences in binding kinetics towards mouse and human FcRn (hFcRn)[28–31]. Specifically, hFcRn binds weakly to mouse IgG (mIgG), whereas mouse FcRn (mFcRn) binds more strongly to hIgG than to mIgG[28,29,31]. The lack of binding of mIgG to hFcRn explains why murine antibodies are rapidly removed from the circulation in humans, despite long half-life in mice[32]. Injected recombinant IgG variants compete with endogenous ligands for FcRn binding, and the strong interaction between mFcRn and hIgG explains why the half-life of hIgG is longer than that of mIgG in WT mice[28,33,34]. Furthermore, both the mouse and human receptors bind more strongly to mouse serum albumin (MSA) than to HSA. Importantly, mFcRn binds HSA very poorly[29], and consequently, HSA has a short half-life in WT mice[35,36]. Mice that are knock-out for mFcRn and transgenic for hFcRn have lower levels of mIgG and higher levels of MSA than what is found in WT mice[21,37]. Due to

the high MSA levels, HSA also have short half-life in these mice[35]. Recently, hFcRn transgenic mice that lack MSA expression have been developed, and injected HSA shows considerably extended half-life of more than 20 days in these mice[36].

Development of engineered IgG and albumin molecules with improved pharmacokinetics requires efficient screening procedures in which FcRn binding and cellular transport can be quantitatively closely monitored[6,7]. A major challenge in Fc-engineering for improved pharmacokinetics is to increase the binding affinity for FcRn at acidic pH without a concomitant increase in affinity at near neutral pH. This is crucial, as low affinity at near neutral pH is a prerequisite for efficient recycling and exocytosis, and one is faced with the same challenge when engineering the FcRn–albumin interaction. Furthermore, such engineering affects binding to the mouse and human receptors differently[28,30], making it even more difficult to make reliable predictions. Except for advanced imaging analysis of molecular sorting and trafficking, no easy and robust in vitro cellular assay is available that allows predictions to be made on how well engineered IgG and albumin variants are rescued from intracellular degradation. Here, we report on a human endothelial cell-based recycling assay, abbreviated HERA, which is established to enable such screening of IgG and albumin variants, prior to evaluation in transgenic mice or non-human primates. Only small amounts of ligands are required, and there is no need for chemical labelling.

## Results

**HERA.** We utilized a previously described human microvascular endothelial cell (HMEC1) line that grows as adherent monolayers. It is modified to stably express human β2-microglobulin and the full-length hFcRn HC with an N-terminal HA-tag and EGFP-fused C-terminally (HMEC1-hFcRn)[38]. The expression of hFcRn was measured to be >100-fold higher than in the parental cell line (Supplementary Fig. 1a–c). We grew the HMEC1-hFcRn cells in wells until 95–100% confluency, removed the medium and incubated the cells for 1 h in HBSS buffer at pH 7.4. Subsequently, fresh buffer at pH 7.4 or pH 6.0 containing the proteins of interest were added, followed by incubation to allow for uptake. We found that an incubation step of 4 h was sufficient (Supplementary Fig. 2a). After washing, the cells were again incubated for 4 h with buffer at pH 7.4 to allow for ligand release (Supplementary Fig. 2b). The amount of ligand that was taken up after the first 4 h incubation and the ligand recycled after additional 4 h was quantified by enzyme-linked immunosorbent assay (ELISA). In addition, we quantified the amounts remaining inside the cells after the recycling step by running ELISA on cell lysates. An illustration of the HERA principle is given in Fig. 1.

To address the effect of extracellular pH on rescue, titrated amounts of IgG were given at either pH 7.4 or pH 6.0 followed by washing and additional 4 h incubation at pH 7.4. The antibody used was a chimeric hIgG1 with mouse lambda light chain, hIgG1 constant HC, and a VH providing specificity for the hapten 4-

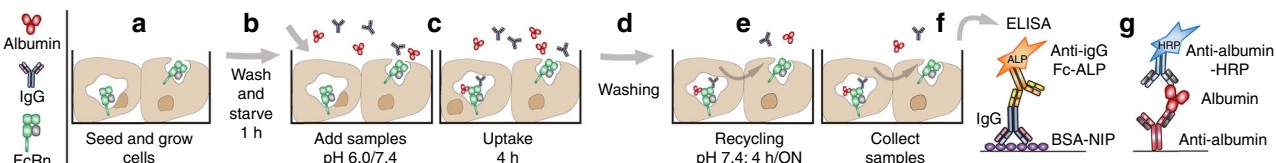

**Fig. 1** The HERA protocol. **a** HMEC1-hFcRn cells are seeded and grown until confluent. **b** Cells are washed and starved for 1 h using HBSS (pH 7.4). **c** The protein of interest is diluted in HBSS (pH 7.4 or 6.0) and added to cells and incubated for 4 h. **d** Medium is removed and cells are extensively washed with ice cold HBSS (pH 7.4) before the cells are lysed or **e** pre-heated HBSS (for 4 h incubation) or growth medium (for overnight incubation (ON)) is added, incubated and samples collected. **f** Cells are extensively washed with ice cold HBSS (pH 7.4) and lysed. **g** The collected samples are analysed in ELISAs specific for IgG or albumin

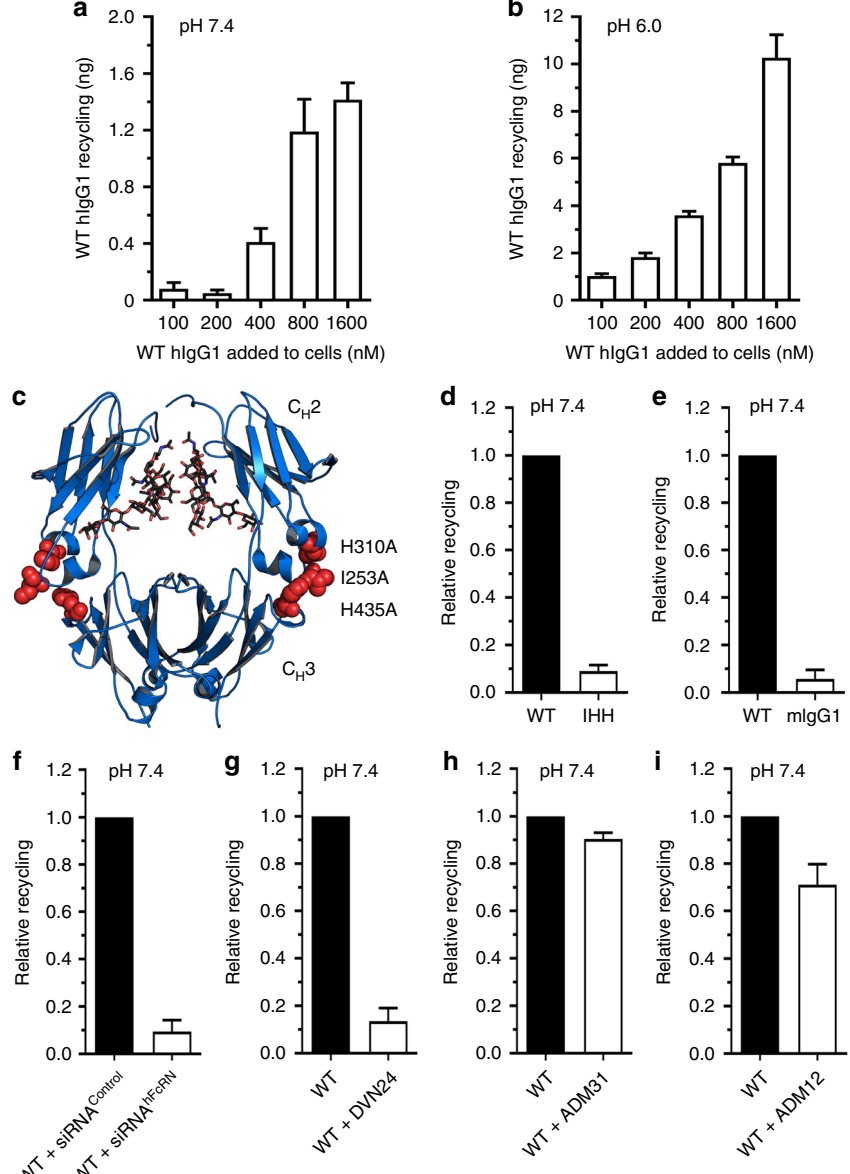

**Fig. 2** Recycling of IgG is dependent on FcRn. Recycling of WT hIgG1 at **a** pH 7.4 and **b** pH 6.0 after 4 h incubation with titrated antibody concentrations (100–1600 nM). **c** An illustration of the hIgG1 Fc crystal structure where the mutated residues in hIgG1-IHH (I253A, H310A and H435A) are highlighted in red. The figure was designed by using the PyMOL software with the crystallographic data of hIgG1 Fc[69]. The amount (ng) recycled for each of the hIgG1 variants was used to calculate relative recycling compared with the WT that was set to 1.0. Histograms showing relative recycling of **d** WT and hIgG1-IHH, **e** WT hIgG1 and mIgG1, and **f** WT hIgG1 by HMEC1-FcRn cells transfected with control or hFcRn-specific siRNAs. Relative recycling of WT hIgG1 in the presence of **g** DVN24 (binds the IgG binding site on hFcRn), **h** ADM31 (binds the HSA binding site on hFcRn) and **i** ADM12 (does not bind the ligand binding sites) are shown. Obtained data are given as mean ± s.d. of three independent experiments performed in duplicates

hydroxy-3-iodo-nitrophenylacetic acid (NIP). The amounts of anti-NIP antibody released from the cells were quantified, and at both pH conditions, concentration-dependent recycling was measured. However, roughly 10-fold more WT hIgG1 was released when the initial incubation step was performed at pH 6.0 (Fig. 2a, b). Thus, acidification during the incubation step sensitizes the assay.

Next, we compared recycling of the WT antibody with that of a hIgG1 variant containing three amino acid residue substitutions in the CH2 and CH3 domains of the Fc (I253A/H310A/H435A; hIgG1-IHH) (Fig. 2c), which eliminate IgG binding to hFcRn[26]. In addition, we included a recombinant anti-NIP WT mIgG1. When equal amounts of the antibodies were added to the cells at pH 7.4, both hIgG1-IHH and mIgG1 were found to be recycled

poorly (Fig. 2d, e). This was as expected, as mIgGs bind only weakly to hFcRn[28,29,31]. Further, we found that HMEC1-hFcRn cells treated with a mixture of siRNAs targeting the hFcRn HC gene downregulated the protein level by nearly 80% for up to 96 h (Supplementary Fig. 3a–b), which resulted in reduced recycling of WT hIgG1 by 90% compared with cells given control siRNA (Fig. 2f). Lastly, cells were incubated with WT hIgG1 in the presence of three monoclonal mouse antibodies raised against hFcRn, which block the binding site for IgG (DVN24), the binding site for albumin (ADM31), or the binding of neither ligand (ADM12)[39,40]. In the presence of DVN24, IgG recycling was abolished, while ADM31 did not interfere and ADM12 gave a minor reduction (Fig. 2g–i). Thus, recycling of hIgG1 was fully dependent on hFcRn.

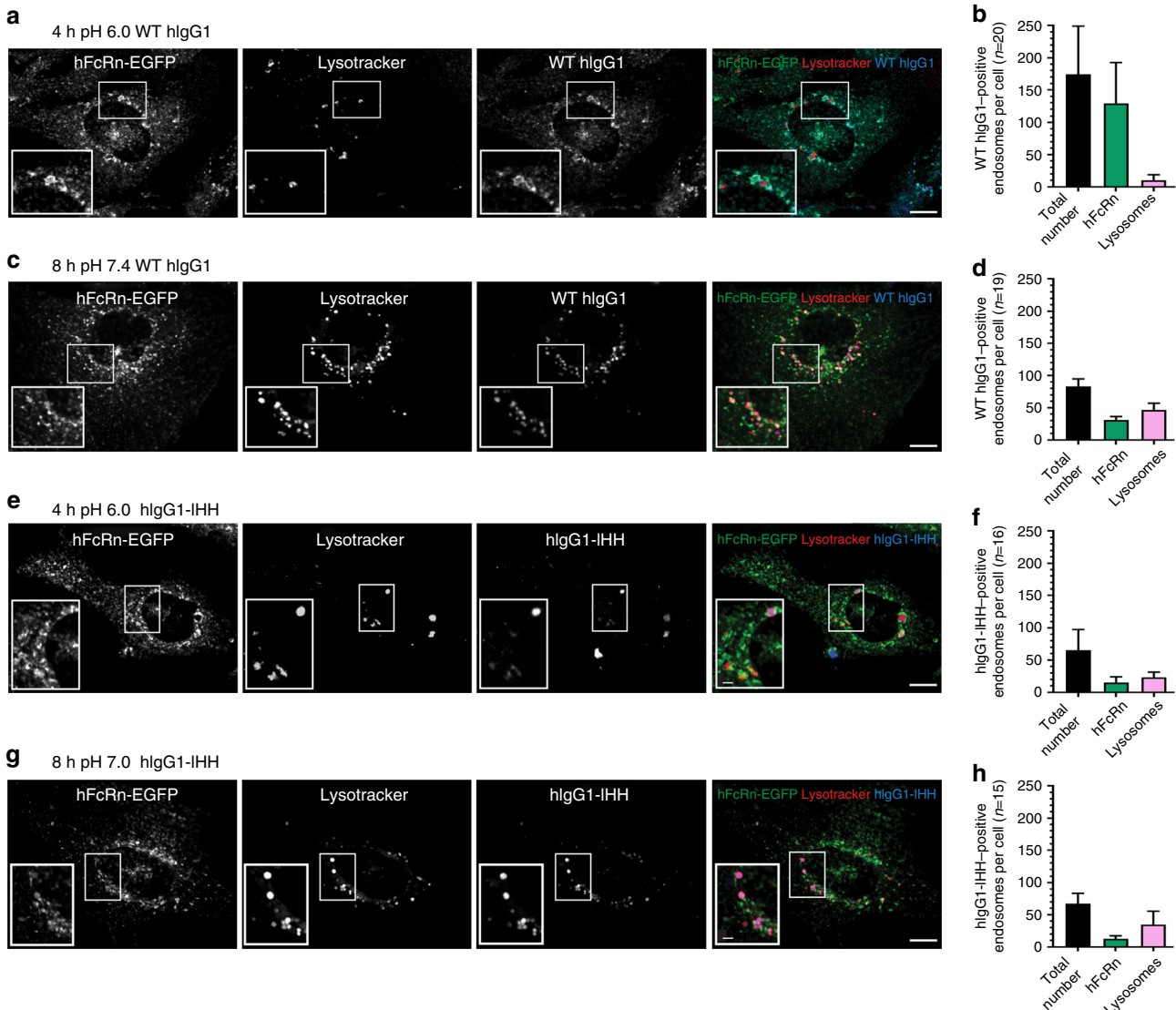

**Fig. 3** Visualization of FcRn-mediated transport of IgG in HMEC1-hFcRn cells. HMEC1-hFcRn cells were seeded in eight-well Nunc™ Lab-Tek™ chambered Coverglass imaging dishes, and live cell imaging was performed the day after. **a** Cells were washed three times with HBSS pH 6.0 and incubated with 400 nM Alexa-labelled WT hIgG1 diluted in HBSS pH 6.0 for 4 h at 37 °C. Cells were incubated with Lysotracker DND-99 for 30 min before washed with HBSS pH 6.0 and images were taken. **b** WT hIgG1 co-localization with FcRn-positive endosomes or lysosomes after 4 h incubation ($n = 20$ cells). **c** The same cells as in **a** were washed three times with HBSS pH 7.4 followed by additional 4 h incubation at 37 °C before images were taken. **d** WT hIgG1 co-localization with FcRn-positive endosomes or lysosomes after additional 4 h incubation ($n = 19$ cells). **e** Cells were washed three times with HBSS pH 6.0 and incubated with 400 nM Alexa-labelled hIgG1-IHH diluted in HBSS pH 6.0 for 4 h at 37 °C. Cells were incubated with Lysotracker DND-99 for 30 min and washed with HBSS pH 6.0 before images were taken. **f** hIgG1-IHH co-localization with FcRn-positive endosomes or lysosomes after 4 h incubation ($n = 16$ cells). **g** The same cells as in **e** were washed three times with HBSS pH 7.4 followed by additional 4 h incubation at 37 °C before images were taken. **h** hIgG1-IHH co-localization with FcRn-positive endosomes or lysosomes after additional 4 h incubation ($n = 15$ cells). Representative confocal images at each condition are shown. Large scale bar—10 μm; small scale bar—2 μm. Co-localization analyses were performed using Imaris spot co-localization. Data are shown as mean ± s.d. from two independent experiments were $n = 20$ (**b**), 19 (**d**), 16 (**f**) and 15 (**g**) cells

**Imaging of uptake and rescue of IgG.** To visualize hIgG1 uptake and release as a function of pH, we performed live cell confocal imaging of HMEC1-hFcRn cells. The cells were grown to 50–70% confluency before Alexa-647-conjugated WT hIgG1 or hIgG1-IHH was added at pH 6.0 and incubated for 4 h. Imaging revealed a high degree of co-localization of WT hIgG1 with the EGFP-fused receptor, suggesting that WT hIgG1 localized to recycling endosomes (Fig. 3a, b). After further incubation at pH 7.4, only low signals for the antibody were observed, suggesting that most of the antibodies had been released to the medium and a minor fraction sorted to the lysosomes (Fig. 3c, d). In contrast, we detected only minor co-localization of hFcRn-EGFP with Alexa-647-conjugated hIgG1-IHH at pH 6.0 whereas a larger fraction of the antibody was co-localized with a lysosomal marker (Fig. 3e, f). After buffer exchange to pH 7.4, even more hIgG1-IHH co-localized with the lysosomal marker (Fig. 3g, h). In addition, the set-up was repeated where both incubation steps were performed at neutral pH and the imaging results revealed that uptake of WT hIgG1 and hIgG1-IHH were similar at pH 7.4 (Supplementary Fig. 4a–d). The major fraction of WT hIgG1 was shown to co-localize with FcRn-positive endosomes (Supplementary Fig. 4b) while most of hIgG1-IHH was localized to lysosomes (Supplementary Fig. 4d).

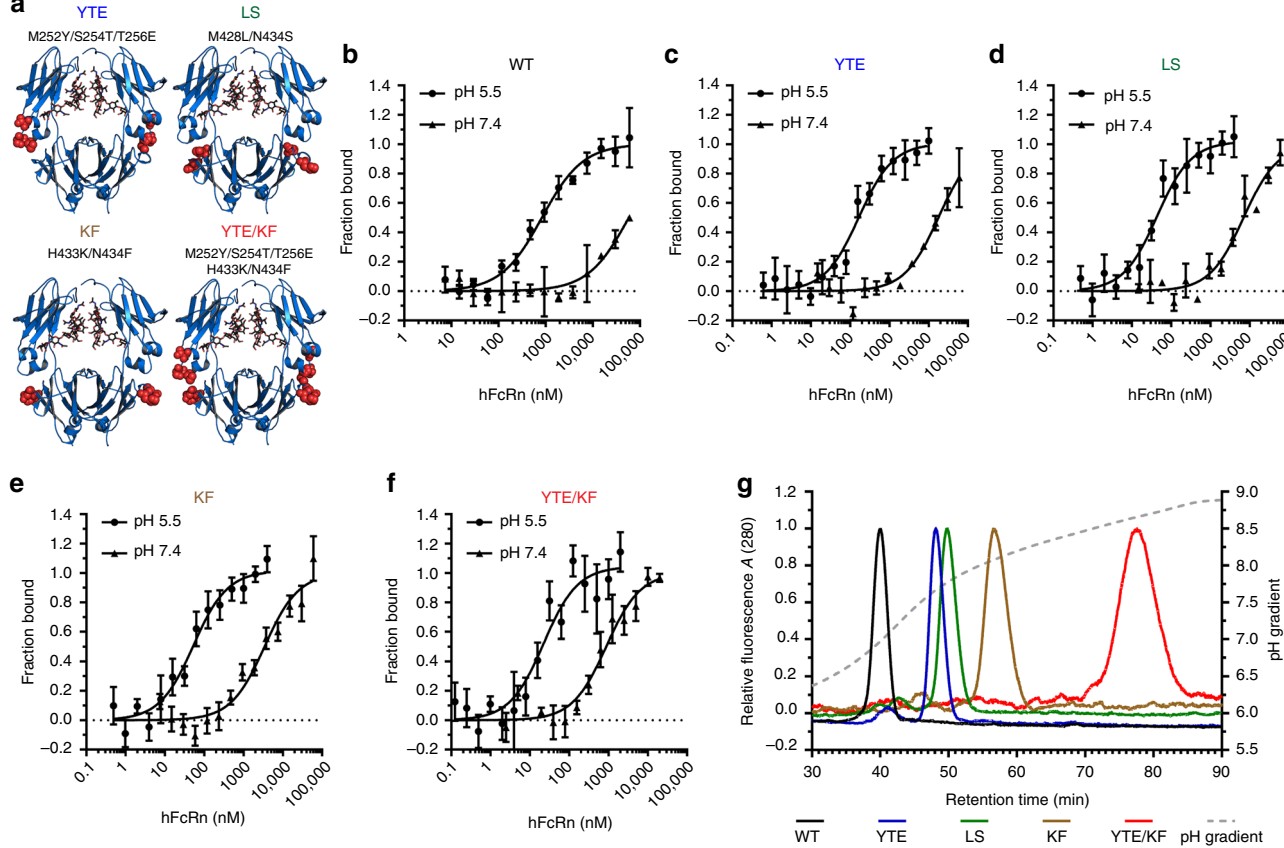

**Fig. 4** hFcRn binding characteristics of Fc-engineered IgG variants. **a** Illustrations of the crystal structure of hIgG1 Fc (blue) with the amino acid residues targeted by mutagenesis highlighted in red spheres: M252Y/S254T/T256E (YTE), M428L/N434S (LS), H433K/N434F (KF) and M252Y/S254T/T256E/H433K/N434F (YTE/KF). The figure was designed using the PyMOL software with the crystallographic data of hIgG1 Fc[69]. MicroScale Thermophoresis analysis where constant amounts (20 nM) of **b** WT hIgG1, **c** hIgG1-YTE, **d** hIgG1-LS, **e** hIgG1-KF and **f** hIgG1-YTE/KF were added to titrated amounts of hFcRn at pH 5.5 or 7.4. Binding data are derived from the specific change in the thermophoretic mobility and the ratio of normalized time-averaged (1 s) fluorescence intensities at defined time points of the MicroScale Thermophoresis traces (−1 and 5 s). The data represent three independent experiments with three replicates; error bars show ± s.d. between independent experiments. **g** Analytical hFcRn affinity chromatography of WT and Fc-engineered hIgG1 variants. The antibody elution profiles are shown as relative fluorescence intensity and as a function of a pH gradient

**FcRn binding properties of Fc-engineered hIgG1 variants.** Next, we compared WT hIgG1 with four previously described hIgG1 variants Fc engineered for altered FcRn-binding kinetics[28,41–43], all with NIP specificity[44]. Two of the antibodies have two substitutions in the CH3-domain, M428L/N434S (hIgG1-LS) and H433K/N434F (hIgG1-KF), the third variant has three substitutions in CH2, M252Y/S254T/T256E and two in the CH3-domain, H433K/N434F (hIgG1-YTE/KF), while the fourth has the three CH2 mutations only (hIgG1-YTE) (Fig. 4a). The binding affinities of purified antibody preparations were determined using MicroScale Thermophoresis at both pH 5.5 and pH 7.4, by adding different concentrations of monomeric hFcRn to a constant amount of labelled hIgG1 variants (Fig. 4b–f). The results obtained at pH 5.5 revealed a hierarchy of FcRn binding relative to WT ($K_D = 0.9 \mu M$) as follows: YTE/KF bound 100-fold more strongly, LS and KF showed similar binding affinities and bound 22- and 18-fold stronger, respectively, while YTE was the weakest binder with a 6-fold increase in affinity. At pH 7.4, the WT bound very weakly (>60 μM) followed by 3-fold stronger binding of YTE, while the LS and KF variants bound approximately 9-fold and 18-fold more strongly than the WT, whereas YTE/KF again was the strongest binder with more than 75-fold enhanced binding (Table 1).

FcRn-mediated recycling is a dynamic process that occurs through a pH gradient. To investigate how Fc-engineering

affected dissociation from hFcRn as a function of pH, we added the antibodies onto an hFcRn-coupled affinity column at pH 6.0 and detected elution as the pH of the buffer was gradually increased towards pH 8.8 (Fig. 4g, Supplementary Table 1). The hIgG1 variants showed distinct profiles, where dissociation of the WT peaked at pH 7.0. YTE, LS and KF eluted with peaks in the pH range of 7.7–8.0. As expected, YTE/KF was released from the column at the highest pH, at 8.5. Thus, the Fc-engineered variants showed distinct pH-dependent binding kinetics as well as pH elution profiles.

**HERA screening of Fc-engineered hIgG1 variants.** HERA was performed on the Fc-engineered hIgG1 variants as described above. We quantified the amounts of hIgG1 variants present inside the cells after 4 h incubation at pH 7.4 and compared with WT. The results showed 5-fold more YTE and LS, 8-fold more KF and 13-fold more YTE/KF (Fig. 5a, d). Quantification of the amounts of antibody released from the cells after washing and the second 4 h incubation with fresh medium showed 6-fold more YTE and LS, 9-fold more KF and 13-fold more of the YTE/KF variant than WT (Fig. 5b, e). Quantification of the amounts that remained in the cells at termination of the assay revealed that 3-fold more YTE and LS, 10-fold more KF and 50-fold more YTE/KF accumulated compared to WT (Fig. 5c, f). The release step could also be run overnight without affecting the relative

**Table 1 KD values derived from MicroScale Thermophoresis analysis**

|  | KD ± s.d. (nM) pH 5.5 | KD ± s.d. (μM) pH 7.4 | Fold difference compared with WT pH 5.5 | Fold difference compared with WT pH 7.4 |
|---|---|---|---|---|
| hIgG1 variants |  |  |  |  |
| WT | 964.3 ± 128.0 | >60 |  |  |
| YTE | 166.4 ± 20.1 | >20 | 5.8 | >3 |
| LS | 43.6 ± 8.3 | 7.0 ± 2.0 | 22.0 | >9 |
| KF | 53.4 ± 10.4 | 3.3 ± 0.9 | 18.0 | >18 |
| YTE/KF | 9.3 ± 5.3 | 0.8 ± 0.2 | 107.0 | >75 |
| HSA variants |  |  |  |  |
| WT | 520.5 ± 96.5 | NA* |  |  |
| K573P | 49.6 ± 8.5 | >70 | 10.0 | NA |

*\* NA not acquired*

differences between the hIgG1 variants and WT (Supplementary Fig. 5a–c).

We then investigated how initial exposure of the antibodies to the cells at pH 6.0 followed by washing and incubation at pH 7.4 affected uptake, release and amount remaining inside the cells (Fig. 5g–i). After the 4 h incubation at pH 6.0, we detected 3-fold more YTE and LS than that of WT inside the cells, as well as 3.5- and 4-fold more KF and YTE/KF, respectively (Fig. 5g). Moreover, 4–5-fold more of all the variants, including YTE/KF, were detected in the medium compared to the WT (Fig. 5h). However, the amounts of the antibodies remaining inside the cells after additional 4 h incubation at pH 7.4 were roughly the same as when all steps were performed at pH 7.4 (Fig. 5c, i). We also treated the cells with a mixture of siRNAs targeting the hFcRn HC to address whether the receptor was required for uptake, and found that it was not crucial for WT hIgG1, but that YTE/KF was less efficiently taken up when the receptor was downregulated (Supplementary Fig. 6a, b). Next, we calculated a so-called HERA score for each of the hIgG1 variants by dividing the relative difference in recycling (Fig. 5e) by the relative difference in residual amount (Fig. 5f). The estimations gave the highest score for YTE and LS while KF gave a similar score as the WT, and YTE/KF showed the lowest (Fig. 6a).

**HERA score correlates with half-life values in hFcRn mice.** To assess whether the HERA score could be used to predict the half-life of WT and the Fc-engineered hIgG1 variants, we measured the serum half-life in hFcRn transgenic mice after injection of equal amounts of antibody. The derived serum pharmacokinetic profiles showed that YTE and LS exhibited increased serum persistence compared with WT, resulting in half-life values of 12.6 and 11.2 vs 8.4 days, respectively (Fig. 6b, Table 2). Further, the KF variant was cleared somewhat faster than the WT with a half-life of 7.6 days while YTE/KF showed the shortest half-life (Fig. 6b, Table 2). Notably, none of the hIgG1 Fc-engineered variants showed increased serum persistence in WT mice (Fig. 6c). The HERA scores were then plotted toward the serum half-lives obtained in WT and hFcRn transgenic mice (Fig. 6d, e). We found a significant correlation with half-life in humanized mice (p-values of 0.0014), but not in WT mice (p-values of 0.2169). The remarkable differences in half-life of the hIgG1 Fc-engineered variants between the mouse strains are reflected by their distinct binding properties for the mouse and human forms of the receptor, where mFcRn binds more strongly to the antibodies but less pH dependently (Supplementary Fig. 7a–d).

**FcRn-mediated recycling of albumin in HERA.** As FcRn does not only bind IgG, but also albumin, we adopted HERA to study recycling of WT HSA at different concentrations at either pH 7.4 or 6.0. Release into the medium was quantified using ELISA, and at both pH conditions, concentration-dependent release was measured. Roughly three-fold more of HSA was detected in the medium when initial uptake was performed at pH 6.0 (Fig. 7a, b). Next, we tested how a mutant variant containing a single-point mutation in domain III (K500A) (Fig. 7c), which reduces binding to hFcRn by 30-fold[14], and also MSA, which binds more strongly than HSA to hFcRn[29,35], were recycled. MSA was recycled more efficiently than HSA, whereas the K500A mutant was poorly recycled compared with the WT (Fig. 7d, e). In addition, down-regulation of the hFcRn HC expression levels with siRNA resulted in a reduction of almost 70% (Fig. 7f), while addition of ADM31 reduced recycling of HSA by nearly 90% (Fig. 7g). Only a modest reduction was shown in the presence of DVN24 or ADM12 (Fig. 7h, i).

Furthermore, we previously reported on an HSA variant with a single amino acid substitution in the C-terminal end of domain III (K573P) (Fig. 7c), which extended the serum half-life from 5.4 to 8.8 days in cynomolgus monkeys[35]. MicroScale Thermophoresis analysis revealed that K573P bound hFcRn 10-fold more strongly than WT at pH 5.5 and only very weakly at pH 7.4 (>70 μM), while WT had no detectable binding at pH 7.4 (Table 1, Fig. 7j, k). Analysis of the elution profiles on the hFcRn-coupled column showed that the WT and the K573P variant dissociated from the receptor at pH 6.4 and 6.9, respectively (Fig. 7l, Supplementary Table 1). Thus, HSA dissociate earlier than that of WT hIgG1 (Fig. 4g, Supplementary Table 1). Using HERA, four-fold more K573P than WT was detected inside the cells after uptake, and also six-fold more was released during the recycling step (Fig. 7m, n) as well as two-fold more remained inside the cells (Fig. 7o). The findings demonstrate that hFcRn expressed by endothelial cells rescues not only hIgG1, but also HSA from degradation and that engineering for enhanced FcRn binding translates into more efficient recycling and release.

**FcRn transport of its two ligands.** In vitro studies have shown that recombinant soluble FcRn binds IgG and albumin to separate binding sites in a non-cooperative manner[12,16,21,22]. However, whether the presence of one ligand interferes with binding and sorting of the other in a cellular context has not yet been addressed. Thus, we studied uptake and release of WT hIgG1 and HSA when excess amounts of the other ligand were added. Equal amounts of WT hIgG1 (anti-adenovirus 5 hexon capsid protein) or WT HSA were added in the presence of rituximab (8 μM) or anti-NIP hIgG1-YTE/KF (2 μM) (Fig. 8a–d). The results revealed that the presence of YTE/KF reduced recycling of WT hIgG1 by 50%, while no reduction was detected when rituximab was added

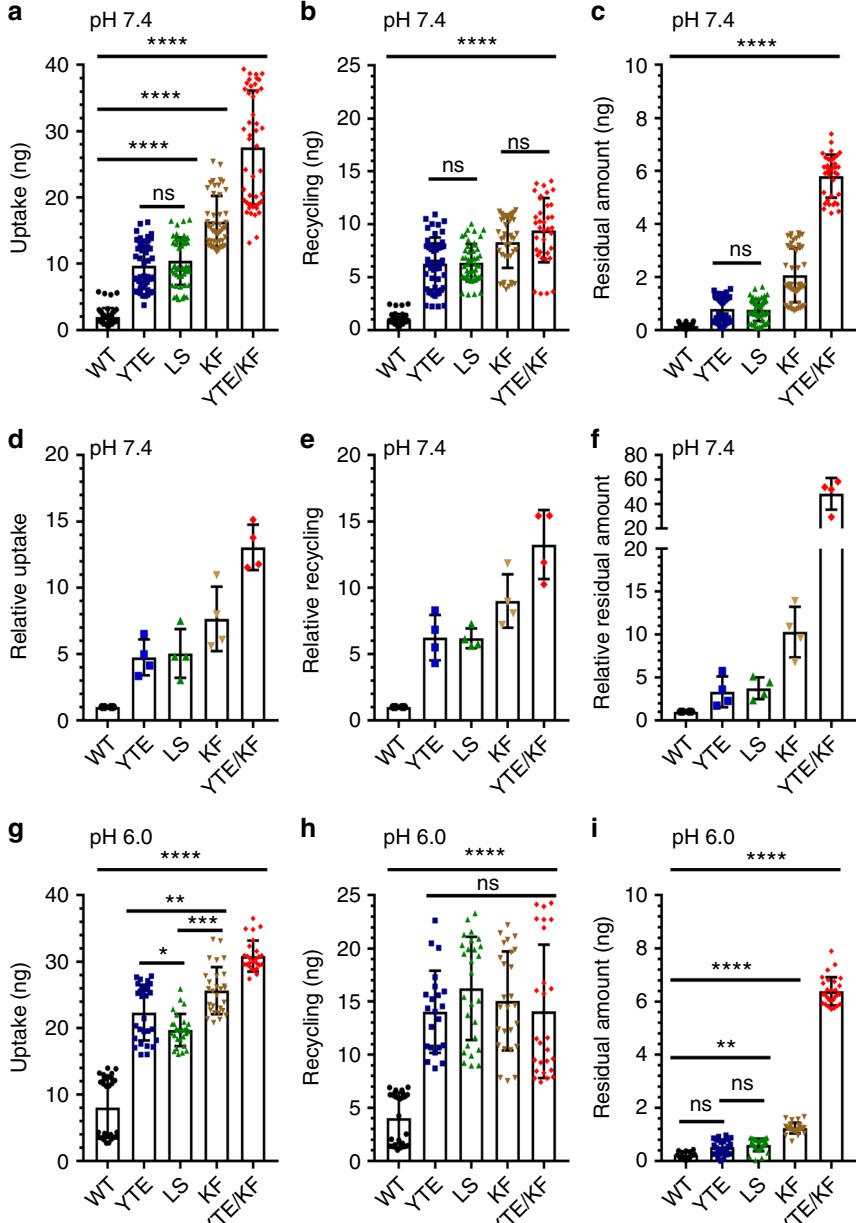

**Fig. 5** HERA screening of WT and Fc-engineered hIgG1 variants. **a** Uptake of WT and Fc-engineered hIgG1 variants at pH 7.4 when 400 nM of each variant was added to the cells followed by 4 h incubation, washing and lysis of the cells. **b** Recycling of the Fc-engineered hIgG1 variants at pH 7.4 when 400 nM of each variant was added to the cells and incubated for 4 h followed by extensive washing and additional 4 h incubation before sample collection. **c** The same procedure as in **b** followed by lysis of the cells. The amounts of hIgG variants in all samples were quantified by ELISA and obtained data are shown as mean ± s.d. of four independent experiments performed in triplicates. ns > 0.05, ****$p < 0.0001$, by one-way ANOVA (Tukey's multiple comparison test). Relative **d** uptake, **e** recycling and **f** residual amount calculated from data (**a–c**), **g** uptake, **h** recycling and **i** residual amounts when WT and the Fc-engineered hIgG1 variants were initially incubated at pH 6.0. The amounts of hIgG variants in all samples were quantified by ELISA, and obtained data are shown as mean ± s.d. of two independent experiments performed in triplicates. ns > 0.05, *$p < 0.05$, **$p < 0.01$, ***$p < 0.001$ and ****$p < 0.0001$, by one-way ANOVA (Tukey's multiple comparison test)

(Fig. 8b). Decreased levels were also detected for WT hIgG1 inside the cells, due to faster degradation expected in the absence of hFcRn available for binding (Fig. 8a), while uptake and rescue of HSA was unaffected except for a slight increase in rescue in the presence of an excess amount of rituximab (Fig. 8c, d).

Furthermore, we measured uptake and recycling of WT anti-NIP hIgG1 in the presence of WT HSA (8 μM) or an engineered HSA variant (E505Q/T527M/V547A/K573P; EQTMVAKP) (2 μM) that binds hFcRn strongly and less pH dependently (Supplementary Fig. 8). While the engineered HSA variant did

not affect uptake, the presence of WT HSA reduced uptake of hIgG1 by roughly 50% (Fig. 8e). Despite this, the amount of hIgG1 taken up was as efficiently recycled, as in the absence of WT HSA, while blocking of the albumin-binding site actually somewhat increased recycling of hIgG1 (Fig. 8f).

## Discussion

Here we describe a cellular assay (HERA) that can be used for rapid screening of hFcRn-dependent rescue from degradation of

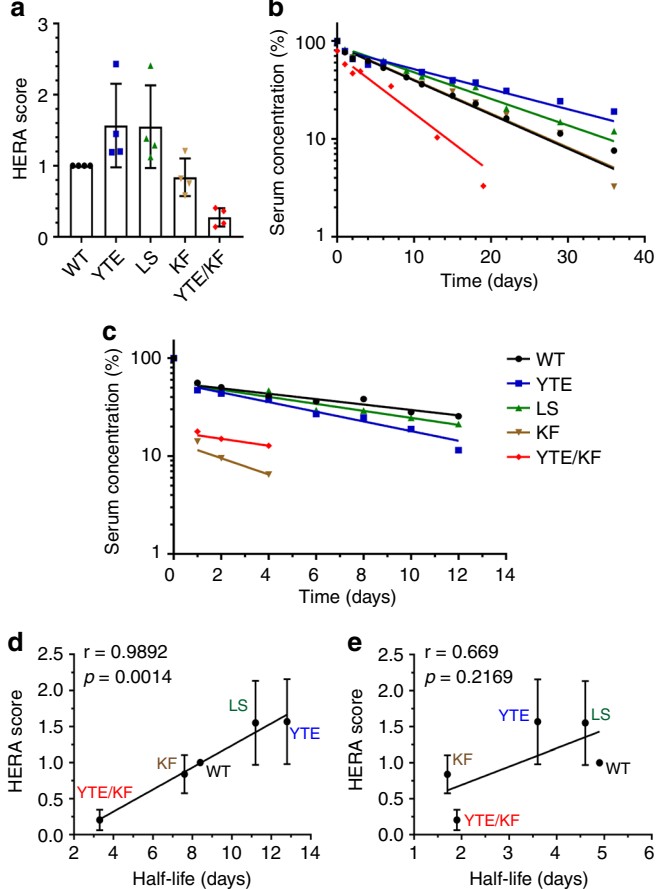

**Fig. 6** HERA score correlates with in vivo serum half-life in hFcRn transgenic mice. **a** HERA score for the WT and the Fc-engineered hIgG1 variants were calculated from the data shown in Fig. 5e, f. **b**, **c** Log-linear changes in the serum concentration of hIgG1 WT (black), YTE (blue), LS (green), KF (brown) and YTE/KF (red) in hFcRn transgenic mice and WT mice, respectively. The antibodies were administrated as a single i.v. injection to 5 mice per group. Correlations of the derived HERA scores with serum half-life values of hIgG1 variants obtained from **d** hFcRn transgenic (r = 0.9892, p < 0.0014) and **e** WT mice (r = 0.669, p < 0.2169). The data are represented as mean ± s.d.

engineered hIgG and HSA variants. We utilized the endothelial cell line HMEC1, which stably expresses hFcRn and can be grown as adherent monolayers, and uniquely regulates rescue of both IgG and albumin from degradation. The HMEC1-hFcRn cell line was chosen, as endothelial cells are known to be major sites for rescue of IgG and albumin from degradation in an FcRn-dependent manner[24,45]. In addition, the cell line has been used extensively to study intracellular FcRn-mediated trafficking of IgG using imaging technologies[17,18,38,46]. The cells were grown as confluent monolayers. FcRn ligands were added and incubated with the cells for 4 h before the medium was removed by washing. During this period, ligands entered the cells, and FcRn binding and recycling or degradation was initiated. However, we lysed cell samples at this time point and defined the amount of ligand found intracellularly as "uptake". In a parallel set-up, fresh medium was instead added to collect ligand that was recycled and released during the subsequent 4 h incubation, defined as "recycling". At this time, these cells were also lysed and the amount of ligand found intracellularly was defined as "residual amount". To normalize the assay, we present the obtained values as relative to that found for the WT ligand. We observed that the recycled and residual amounts did not equal the amount taken up. This is

likely due to degradation, a notion that is also supported by live imaging showing a fraction of IgG in the lysosomes at the end of experiments.

We used the assay to characterize and compare a panel of hIgG1 variants, with the same framework and specificity for NIP. All contained well-characterized Fc mutations that have been reported to improve FcRn binding and give rise to extended half-life in mice, non-human primates and humans[28,41–43,47]. In addition, we included a so-called Abdeg molecule (antibody that enhances degradation) that is engineered to bind more strongly in the pH range of 6.0–7.4[41,48]. The IgG variants were added to the medium at pH 6.0 or 7.4. While considerably more of YTE, LS and KF were taken up and detected in the medium after recycling compared with the WT, no major differences were detected between these Fc-engineered variants. In contrast, cellular accumulation measured as residual amount varied in that more KF was found inside the cells. This corresponds well with the fact that KF binds more strongly to FcRn at pH 7.4 and has a longer retention time on the FcRn column. Comparing LS and YTE, LS bound hFcRn almost four times more strongly than YTE at pH 5.5. However, it was not found in higher amounts inside the cells at the start of the recycling step, nor was it more efficiently recycled. This could be explained by stronger receptor binding at neutral pH, but the difference in retention time between the variants on the hFcRn-coupled column is minor.

Furthermore, we determined the half-life of the panel of anti-NIP hIgG1 variants in WT and hFcRn transgenic mice and measured profound differences in the two strains, which mirror their distinct binding properties toward the mouse and human forms of FcRn[28–31]. Again, this highlights the importance of considering cross-species differences when measuring pharma-cokinetics. Importantly, the calculated HERA scores for the hIgG1 variants were shown to correlate with in vivo half-life in hFcRn transgenic mice, but not WT mice. The assay is fast and convenient and we find it extremely useful and also sensitive. Furthermore, we found that the sensitivity could be increased by doing the initial uptake step at pH 6.0.

FcRn is predominantly found to be expressed within endosomal compartments, while only a minor fraction is displayed at or near the cell surface at least in parenchymal cells[18,19]. The influence of pH on FcRn-mediated uptake and trafficking of IgG in different types of cells is a matter of debate[49,50]. Due to negligible FcRn binding of IgG at neutral pH, cellular uptake may solely be dependent on fluid-phase pinocytosis, but data also support active involvement of FcRn[51]. The Fc-engineered variants with increased binding at pH 7.4 may thus engage FcRn at the cell surface or in early endosomes more efficiently than the WT. Our findings show that more of the WT and Fc-engineered variants were detected inside the cells when incubation was done at pH 6.0. This may support the notion above, but incubation at acidic pH will also prevent release during exocytosis and as such promote accumulation inside the cells. This is also supported by imaging, where the WT and IHH variants were taken up in equal amounts when given to the cells at pH 7.4. More of the WT was taken up than IHH when the assay was performed at pH 6.0. In line with this, when cells were treated with siRNA targeting FcRn, we found that uptake of WT hIgG1 was not dependent on the receptor, while active FcRn-mediated uptake was measured for the YTE/KF mutant variant. As FcRn is expressed as a fusion with a C-terminal EGFP, HERA may be combined with imaging studies to gain further mechanistic insights into how FcRn binds and transports therapeutics.

We found the KF mutant to have a similar HERA score and half-life as WT hIgG1 in hFcRn transgenic mice. It bound hFcRn with 3.3 μM affinity at neutral pH, while LS, which was more efficiently rescued, bound hFcRn with an affinity of 7 μM at this

**Table 2 Serum half-life data in WT and humanized FcRn mice**

| IgG1 variants | IgG1 specificity | Route | Dose (mg/kg) | Half-life (days) ($n = 5$) | ±s.d. (days) | Mouse strains |
|---|---|---|---|---|---|---|
| WT | | | | 8.4 | 2.5 | |
| YTE | | | | 12.6 | 1.2 | |
| LS | anti-NIP | IV | 5 | 11.2 | 2.2 | Tg32 hemizygous mice |
| KF | | | | 7.6 | 1.0 | |
| YTE/KF | | | | 3.3 | 0.3 | Tg32 homozygous mice |
| WT | | | | 4.9 | 0.4 | |
| YTE | | | | 3.6 | 0.6 | |
| LS | anti-NIP | IV | 5 | 4.6 | 1.2 | WT mice |
| KF | | | | <2.0 | | |
| YTE/KF | | | | <2.0 | | |

pH. This means that the affinity threshold to achieve efficient recycling and release lies between 3–7 μM as defined by Micro-Scale Thermophoresis. Notably, it has previously been shown that high affinity at acidic pH combined with binding affinity up to 0.8 μM at neutral pH, as defined by surface plasmon resonance measurements, resulted in extended half-life[52].

While relative uptake and rescue of the three hIgG1 variants, YTE, LS and KF, were similar at the two pH conditions, this was not the case for YTE/KF, which was distinctly different regarding uptake, recycling and residual amount remaining inside the cells. The amount of YTE/KF recycled represented a minor fraction of the total pool of YTE/KF that was taken up and accumulated inside the cells, which is in accordance with live-imaging studies[41,46]. The distinct distribution of YTE/KF is mirrored by its nM affinity at pH 5.5, more than 100-fold stronger than the WT, and sub-μM affinity at pH 7.4, roughly 10-fold weaker than at pH 5.5. Our study thus demonstrates that both the amounts recycled and remaining inside the cells should be measured to make a useful prediction of how hFcRn-expressing cells will sort ligand variants in vivo. Thus, the residual amount of ligands was taken into consideration when calculating the HERA score, as well as the amount recycled.

We also adopted HERA to study HSA variants, and again, recycling was shown to depend on hFcRn, as only WT HSA and not a null binding mutant was rescued. Further, a HSA variant (K573P) engineered for improved pH-dependent FcRn binding showed enhanced recycling and a HERA score of 2.5, which mirrors its extended half-life in hFcRn transgenic mice and cynomolgus monkeys[35].

Although it is well-established that both IgG and albumin bind FcRn in a similar pH-dependent manner[11–14], we found a distinct difference in dissociation as WT HSA was shown to elute earlier from the hFcRn column (pH 6.5) than WT hIgG1 (pH 7.0). This finding may shed new light on how the two ligands are transported in different cell types through a pH gradient. With regard to engineering for enhanced rescue, it suggests that "the window for improvement" is wider for HSA than for IgG. Moreover, we showed that the ligands are transported by FcRn in the absence of the other, and the presence of rituximab did not negatively affect HSA rescue, but instead had a slightly enhancing effect.

Studies in mice have shown that the levels of endogenous IgG are considerably reduced in the presence of YTE/KF[41,46,48]. We therefore used the YTE/KF variant to study transport of hIgG1 and HSA in HERA. In the presence of excess amounts of YTE/KF, where most of the IgG binding sites on hFcRn are expected to be occupied, reduced rescue of WT anti-hexon hIgG1 was observed, which was not observed in the presence of rituximab. In contrast, rescue of HSA was not affected. This pin points that HSA is still recycled and released when IgG is bound to the

receptor, which is in line with data showing that YTE/KF is not affecting receptor trafficking, but instead hijacks the receptor throughout the endosomal pathway[41,53]. When hIgG1 was given in the presence of an engineered HSA variant that binds strongly and less pH dependently, the antibody was recycled somewhat more efficiently than in the presence of WT HSA. Notably, while uptake of hIgG1 was unaffected in the presence of the engineered albumin variant, the presence of WT HSA reduced uptake of the antibody. Whether this reflects biology or is an artefact due to the large amounts of WT HSA added needs to be further investigated.

In addition to the direct interaction between FcRn and its ligands, there are several additional factors that may contribute to define half-life, such as the expression level and availability of the target antigen, which may enhance clearance. Thus, HERA is expected to only correlate with half-life when the clearance mechanism is studied at dose levels dominated by FcRn binding and transport. However, HERA should be an attractive tool to study how the presence of soluble antigen is affecting cellular transport of the immune complex as well as how engineering for pH-dependent antigen binding is affecting recycling and release of the ligand and antigen, respectively[54,55].

Further, it was recently shown that the pI of IgG may influence cellular uptake[55]. In addition, the choice of framework can affect the pI, and thus the half-life[56]. It is also a matter of debate whether the Fab arms may influence FcRn binding. Support for this has been provided using hydrogen–deuterium exchange mass spectrometry, which indicated that there may either be a direct interaction or possibly a conformational link between the Fab arms and the constant Fc during complex formation with the receptor[57]. A direct role of the Fab arms has been shown to depend on the charge of the Fv part, as IgGs with differences in charge have a great influence on pH-dependent dissociation from an hFcRn-coupled column[58] and receptor-binding kinetics[59]. The effect of such structural differences on uptake and FcRn-mediated sorting may be addressed using HERA. However, during early stage development, it is important to eliminate antibodies with off-target reactivity. This may be done using assays assessing binding to panels of non-cognate proteins[60–62]. Interestingly, it was recently shown that this may simply be done by screening the antibodies towards recombinant chaperone proteins that correlated with clearance in mice[63]. After such exclusion, HERA may be an attractive tool for screening and ranking not only monoclonal IgG antibodies and Fc-engineered variants, but also other formats such as anti-FcRn blocking reagents, Fc and albumin fusions as well as immunoconjugates prior to in vivo studies.

## Methods
**Cell culture**. HEK293E cells (ATCC, Manassas, VA, USA) and the J558L murine myeloma cell line stably producing NIP-specific hIgG1-IHH were cultured in

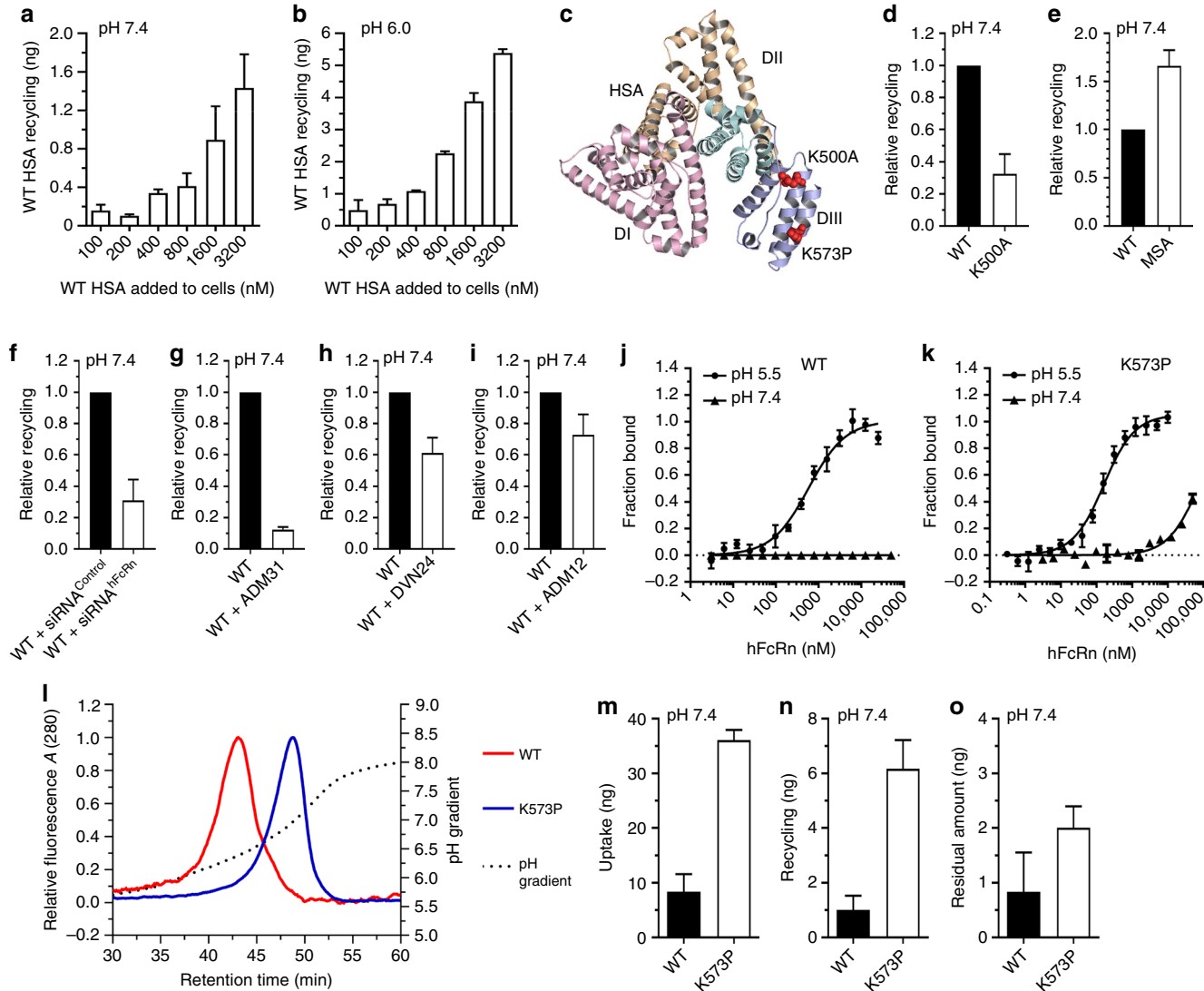

**Fig. 7** HERA screening of HSA variants. Recycling of WT HSA at **a** pH 7.4 and **b** pH 6.0 after 4 h incubation with titrated HSA concentrations (100–3200 nM). **c** An illustration of the crystal structure of HSA with the amino acid residues targeted by mutagenesis (K500A and K573P) highlighted in red spheres. The figure was designed using the PyMOL software with the crystallographic data of HSA[70]. Histograms showing relative recycling of **d** WT HSA and K500A, **e** HSA and MSA, **f** WT HSA by HMEC1-FcRn cells transfected with a control or hFcRn-specific siRNAs. Relative recycling of WT HSA in the absence or presence of either **g** ADM31 (binds the HSA binding site on hFcRn), **h** DVN24 (binds the IgG binding site on hFcRn) or **i** ADM12 (does not bind the ligand binding sites). MicroScale Thermophoresis measurements where **j** WT HSA-EGFP and **k** HSA-K573P-EGFP at a constant concentration (100 nM) were added to titrated amounts of hFcRn at pH 5.5 and 7.4. Binding data were derived from the specific change in thermophoretic mobility and the ratio of normalized time-averaged (1 s) fluorescence intensities at defined time points of the MicroScale Thermophoresis traces (−1 and 4 s). The data from three independent experiments with three replicates are shown where error bars show ± s.d. **l** Analytical hFcRn affinity chromatography of WT and K573P. The elution profiles are shown as relative fluorescence intensity and as a function of a pH gradient. **m** Uptake, **n** recycling and **o** residual amounts when equal amounts of WT and K573P were added at pH 7.4. The amounts of HSA variants were quantified using ELISA. The obtained data are shown as mean ± s.d. of two independent experiments performed in triplicates

RPMI 1640 (Sigma-Aldrich) supplemented with 10% heat-inactivated fetal calf serum (FCS) (Sigma-Aldrich), 2 mM L-glutamine, 25 µg/ml streptomycin, and 25 U/ml penicillin (all from Bio Whittaker). The parental human microvascular endothelial cell line (WT HMEC1) and HMEC1 stably expressing HA-hFcRn-EGFP (HMEC1-hFcRn)[38] were grown in MCDB 131 medium (Gibco) supplemented with 10% heat-inactivated FCS, 2 mM L-glutamine and 25 µg/ml streptomycin, and 25 U/ml penicillin, 10 ng/ml mouse epidermal growth factor (PeproTech) and 1 µg/ml hydrocortisone (Sigma-Aldrich). Medium for HMEC1-hFcRn was also supplemented with 5 µg/ml blasticidin (InvivoGen) and 100 µg/ml G418 (Sigma-Aldrich) to maintain stable expression of hFcRn. High five cells (Invitrogen) were grown in Express FIVE SEF medium (Invitrogen) supplemented with 18 mM L-glutamine and 1% antibiotic–antimyotic (Invitrogen). All cell lines were negative for mycoplasma contamination (MycoAlert™ PLUS Mycoplasma detection kit, Lonza).

**Production of hIgGs**. Vectors encoding the WT and mutated hIgG1 HC variants with specificity for NIP and adenovirus 5 hexon have previously been reported[44,64,65]. The vectors were transiently transfected into HEK293E cells using Lipofectamine 2000 as described by the manufacturer (Life Technologies). Cells were co-transfected with plasmids encoding light chains with the corresponding specificity[64,65]. Production of an anti-NIP hIgG1-IHH variant was done using a previously described J558L murine myeloma cell line stably transfected with pLNOH2-[NIP]hIgG1-IHH, which also constitutively expresses the anti-NIP mouse λ light chain[26]. Growth medium was harvested and replaced every second day for 2 weeks prior to purification using a CaptureSelect™ pre-packed anti-hIgG-CH1 column (Life Technologies) as described by the manufacturer. The collected proteins were up-concentrated and buffer-changed to phosphate-buffered saline (PBS) (Sigma-Aldrich) using Amicon Ultra-15 ml 50K columns (Millipore) prior to size exclusion chromatography using a Superdex 200 increase 10/300GL column (GE

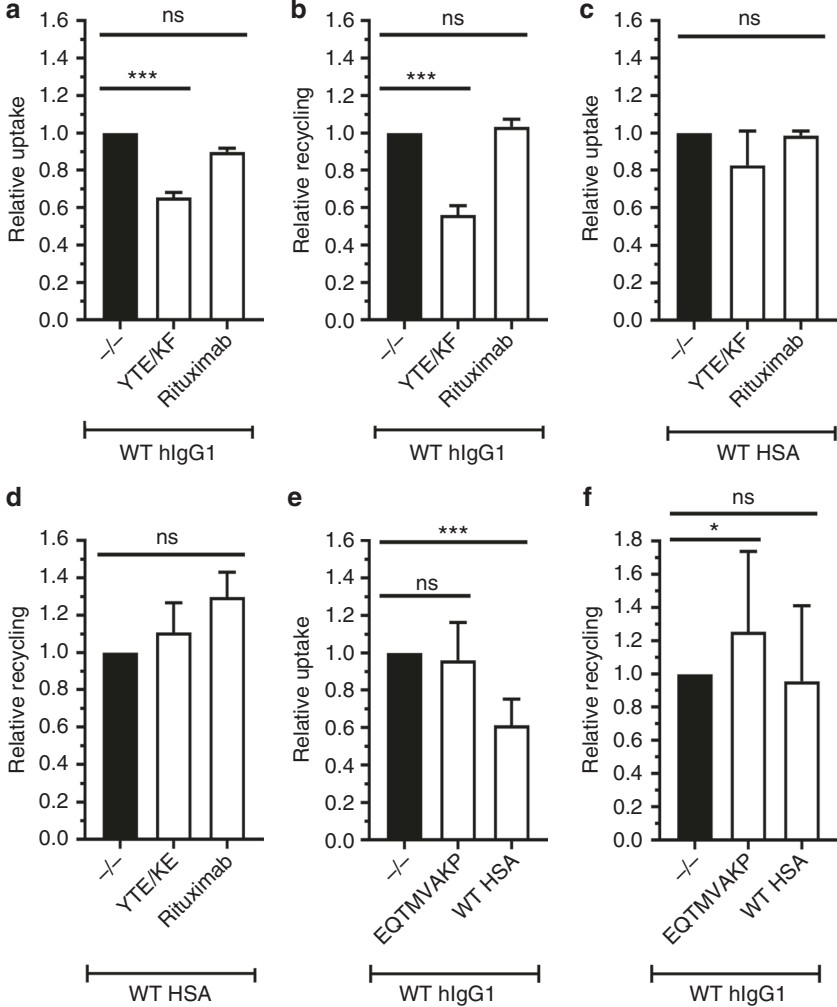

**Fig. 8** YTE/KF is inhibiting rescue of IgG from degradation but not albumin. Relative **a** uptake and **b** recycling of WT hIgG1 in the absence or presence of excess amounts of YTE/KF or rituximab after 4 h incubation at pH 7.4. **c** Relative uptake and **d** recycling of WT HSA in the absence or presence of excess amounts of YTE/KF or rituximab after 4 h incubation at pH 7.4. **e** Relative recycling and **f** uptake of WT hIgG1 in the absence or presence of excess amounts of EQTMVAKP or WT HSA after 4 h incubation at pH 7.4. The amounts of WT hIgG1 and WT HSA were quantified using ELISA. Obtained data are shown as mean ± s.d. of **a**–**d** three and **e**, **f** six independent experiments performed in **a**–**d** duplicates and **e**, **f** triplicates. ns > 0.05, *$p < 0.05$, ***$p < 0.001$, by one-way ANOVA (Dunnett's multiple comparisons test)

Healthcare) and an ÄKTA FPLC instrument (GE Healthcare). The monomeric fractions were up-concentrated by Amicon Ultra-0.5 ml 100K columns (Millipore).

**Production of HSA variants.** The cDNA fragment of full-length HSA was cloned into a pcDNA3.1 (Invitrogen), and cDNA fragments encoding DIII variants (K573P, K500A and E505Q/T527M/V547A/K573P; EQTMVAKP) were ordered (GenScrip, NJ, USA), and sub-cloned into pcDNA3.1-HSA using the restriction sites of *Bam*HI and *Xho*I. In addition, cDNA sequences encoding WT HSA and K573P with a C-terminal glycine-serine linker ((GGS)4GG) were ordered (Gen-Script) and sub-cloned in a frame of a gene encoding EGFP in pEGFP-N1 (Clontech Laboratories) using the restriction sites of *Xho*I and *Hin*dIII. All HSA variants were produced in adherent HEK293E cells by transient transfection using Lipofectamine 2000 (Life Technologies). Growth medium was harvested and replaced for up to 2 weeks prior to purification using CaptureSelect HSA affinity matrix (Life technologies) packed in a 5 ml column (Atoll). Elution from the column was done using 20 mM Tris, 2 M MgCl$_2$ pH 7.4. Collected proteins were up-concentrated and buffer-changed to PBS (Sigma-Aldrich) using Amicon Ultra-15 ml 30K columns (Millipore) prior to size exclusion chromatography using a Superdex 200 increase 10/300GL column (GE Healthcare) and an ÄKTA FPLC instrument (GE Healthcare). Monomeric fractions were up-concentrated by Amicon Ultra-0.5 ml 30K columns (Millipore).

**Production of recombinant hFcRn.** Truncated monomeric His-tagged hFcRn was produced using a Baculovirus expression system[39,66]. The viral stock was a kind gift from Dr. Sally Ward (University of Texas, Southwestern Medical Center, Dallas, USA). Briefly, high five cells were cultured at 1 × 10$^6$ cells/ml at 27 °C with gentle

agitation (160 rpm), before cells (500 ml, 1 × 10$^6$ cells/ml) were infected with 1 ml virus stock (*Autographica californica* nuclear polyhedrosis virus harbouring the pAxUW51 plasmid encoding cDNA of the three extracellular domains of hFcRn HC fused with a His$_6$-tag and human β2-microglobulin). After 72 h at 24 °C post infection, the supernatant was harvested, and the receptor was purified using a HisTrap HP column supplied with Ni2+ ions (GE healthcare). After column equilibration with PBS containing sodium azide (0.05%), the supernatant was adjusted to pH 7.2 and applied on the column with a flow rate 5 ml/min, then the column was washed with 200 ml PBS followed by 50 ml PBS containing 25 mM imidazole pH 7.3, and bound hFcRn was eluted with 50 ml PBS containing 250 mM imidazole. Amicon Ultra 10K columns (Millipore) were used for up-concentration and buffer-change to PBS (Sigma-Aldrich) prior to size exclusion chromatography using a HiLoad 26/600 Superdex 200 prep grade column (GE Healthcare) coupled to an ÄKTA FPLC instrument (GE Healthcare). Monomeric receptor was up-concentrated by Amicon Ultra-0.5 ml 10K columns (Millipore) and stored at 4 °C.

**HERA.** 7.5 × 10$^5$ HMEC1 cells stably expressing HA-hFcRn-EGFP were seeded into 24-well plates per well (Costar) and cultured for 2 days in growth medium. The cells were washed twice and starved for 1 h in Hank's balanced salt solution (HBSS) (Life Technologies). Then, 400 nM of either hIgG1, mIgG1 or 1600 nM of WT HSA and MSA (Calbiochem) were diluted in 250 μl HBSS (pH 7.4) and added to the cells followed by 4 h incubation. The medium was removed and the cells were washed four times with ice cold HBSS (pH 7.4), before fresh warm HBSS (pH 7.4) or growth medium without FCS and supplemented with MEM non-essential amino acids (ThermoFisher) was added. Samples were collected at 4 h or ON. For competition studies, 800 or 3200 nM of DVN24, ADM12 and ADM31[39,40] were diluted

into 200 μl and added to the cells followed by 30 min incubation. The medium was removed and 400 nM anti-NIP hIgG1 or 1600 nM HSA together with DVN24, ADM12 and ADM31 were added. Five hundred nM anti-hexon hIgG1 and WT HSA were mixed with 2 μM anti-NIP hIgG1-YTE/KF or 8 μM rituximab (Roche) in 250 μl HBSS (pH 7.4) and added to the cells. The following steps were performed as described above.

The amounts of anti-NIP or anti-hexon hIgG1 or HSA variants were quantified using ELISA as described below. HERA was also performed where the initial step was done at pH 6.0 using HBSS containing 7 mM MES buffer (Sigma-Aldrich), 200 nM of hIgG1, and the following steps as described above. Total protein lysates were obtained using the CelLytic M cell lysis Reagent (Sigma-Aldrich) or RIPA lysis buffer (ThermoFisher) supplied with a protease inhibitor cocktail (Sigma-Aldrich) or complete protease inhibitor tablets (Roche). The mixture was incubated with the cells on ice and a shaker for 10 min followed by centrifugation for 15 min at $10,000 \times g$ to remove cellular debris. Quantification of the amounts of hIgG1 or HSA variants present in the lysates was done by ELISA.

The derived values for recycling and residual amount for each of the IgG1 variants were used to calculate the HERA score. The following formula was used: $(R_X/R_{WT})/(RA_X/RA_{WT})$. The parameter $R$ (in ng) is recycling at a given time while RA (in ng) is the residual amount. X is the protein of interest while WT is the parental variant used to standardize results.

**Quantification of IgG.** Ninety-six well plates (Costar) were coated with 1 μg/ml of bovine serum albumin (BSA) conjugated to NIP (BSA-NIP) (Biosearch Technologies, Inc.), recombinant AdV5 hexon (AbD Serotec) or polyclonal anti-human or mouse Fc-specific antibody from goat (locally produced) diluted (0.2 μg/ml) in PBS. After incubation ON at 4 °C, plates were blocked with PBS containing 4% skimmed milk (S) (Sigma-Aldrich) for 1 h at room temperature (RT) and washed four times with PBS containing 0.05% Tween 20 (T) (Sigma-Aldrich). Serial dilutions of IgGs (250.0–0.1 ng/ml) were added in parallel with samples collected from HERA and incubated for 2 h at RT. The plates were washed four times as above, before alkaline phosphatase-conjugated polyclonal anti-mIgG or hIgG Fc-Ab from goat (#A2429, #A9544, Sigma-Aldrich), diluted (1:4000) in 4% PBS/S/T, was added for 1.5 h at RT. The plates were washed as above before 100 μl of the p-nitropenylphospate substrate (Sigma-Aldrich) diluted to 10 μg/ml in diethanolamine buffer was added. The absorbance was measured at 405 nm using a Sunrise spectrophotometer (TECAN).

**Quantification of albumin.** Ninety-six well ELISA plates (Costar) were coated with 1.0 μg/ml of polyclonal goat anti-HSA (#A1151, Sigma-Aldrich) or polyclonal anti-MSA (#ab19194, Abcam) antibodies, 1.0 μg/ml diluted in PBS, and incubated ON at 4 °C. Plates were blocked with PBS/S ON at 4 °C and washed four times with PBS/T before serial dilutions of MSA or HSA (250.0–0.1 ng/ml) in PBS/S/T were applied in parallel with samples from HERA followed by 2 h incubation at RT. Bound MSA and HSA were detected using horseradish peroxidase-conjugated polyclonal anti-MSA antibody from goat (#ab19195; Abcam, 1:4000) or alkaline phosphatase-conjugated polyclonal anti-HSA antibody from goat (#A80229AP; Bethyl Laboratories, Inc., 1:5000). ELISAs were developed by adding 100 μl of 3,3´,5,5´-tetramethylbenzidine solution (Calbiochem) and the reaction was stopped by adding 100 μl of 1 M HCl for MSA, while for HSA 100 μl of the p-nitropenyl-phospate substrate (Sigma-Aldrich) diluted to 10 μg/ml in diethanolamine buffer was added. The absorbance was measured at 450 or 405 nm using a Sunrise spectrophotometer (TECAN).

**Quantification of hFcRn.** Membrane-bound proteins were extracted from WT HMEC1 and HMEC1-hFcRn cell lines using a Native Membrane Extraction kit as described by the manufacturer (Calbiochem). The final fractions were stored at −80 °C until analysed in ELISA. Ninety-six-well plates (Costar) were coated with 8 μg/ml of hIgG1-YTE/KF and incubated overnight at 4 °C before plates were blocked with PBS/S for 1 h at RT and then washed four times with PBS/T at pH 5.5 (100 mM phosphate buffer, 0.15 M NaCl, 0.05% Tween 20). Serial dilutions of hFcRn (500.0–0.2 ng/ml) in PBS/T pH 5.5 were added in parallel with 2 μg of extracted membrane proteins followed by incubation for 2 h at RT. The plates were washed four times as above, before 1 μg/ml of biotinylated ADM31 was added and incubated for 1.5 h at RT followed by washing and detection using alkaline phosphatase-conjugated streptavidin (GE healthcare) diluted (1:3000) in PBS/T pH 5.5. The plates were washed as above before 100 μl p-nitropenylphosphate (10 μg/ml) diluted in diethanolamine buffer was added. The absorbance was measured at 405 nm using a Sunrise spectrophotometer (TECAN).

**SDS-PAGE and western blotting.** Extracted membrane protein fractions from HMEC1-hFcRn cells treated with a mixture of control siRNA or siRNA targeting the hFcRn HC were separated on a non-reducing 12% Bis-Tris plus gel (Invitrogen) and transferred onto a polyvinylidene fluoride membrane (Millipore) in Tris/glycine buffer (25 mM Tris, 192 mM glycine, and 20% methanol, pH 8.3) at 25 V for 30 min using a semi-dry blotting apparatus (Bio-Rad). The membrane was blocked with PBS/S/T before added horseradish peroxidase-conjugated anti-HA-tag polyclonal Ab produced in goat (#ab1265, Abcam, 1:5000) to detect HA-tagged hFcRn. The membrane was washed four times with PBS/T and developed with a

SuperSignal West Femto substrate (Pierce) and image acquired by the G:BOX instrument (Syngene).

**FcRn-IgG binding ELISA.** Ninety-six-well plates (Costar) were coated with 2 μg/ml of BSA-NIP (Biosearch Technologies, Inc.) diluted in PBS and incubated ON at 4 °C. Plates were blocked with PBS/S for 1 h at RT and then washed four times with PBS/T. Serial dilutions of WT IgG1 and Fc-engineered variants (10,000–1.5 ng/ml) in PBS/S/T were added and incubated for 1 h at RT. The plates were washed four times as above, before site-specific biotinylated mFcRn and hFcRn (2.5 μg/ml) (Immunitrack) diluted in PBS/S/T pH 5.5 (100 mM phosphate buffer, 0.15 M NaCl, 4% skimmed milk, 0.05% Tween 20) or PBS/S/T pH 7.4 was added followed by incubation for 1 h at RT. After washing four times with either pH 5.5 or pH 7.4 PBS/T, alkaline phosphatase-conjugated streptavidin (GE Healthcare) diluted (1:3000) in pH 5.5 or pH 7.4 PBS/S/T was added for 1 h at RT. The plates were washed as above before 100 μl of the p-nitropenylphospate substrate (Sigma-Aldrich) diluted (10 μg/ml) in diethanolamine buffer was added. The absorbance was measured at 405 nm using a Sunrise spectrophotometer (TECAN).

**FcRn-HSA binding ELISA.** Ninety-six-well ELISA plates (Costar) were coated with 8 μg/ml of hIgG1-YTE/KF diluted in PBS, and incubated ON at 4 °C. Plates were blocked with PBS/S for 1 h at RT, and then washed four times with PBS/T. His-tagged hFcRn (10 μg/ml) in pH 5.5 or pH 7.4 PBS/S/T was added and incubated for 1 h at RT. The plates were washed four times with either pH 5.5 or pH 7.4 PBS/T, before serial dilutions (15,000–7.3 ng/ml) of WT HSA and HSA-EQTMVAKP in pH 5.5 or pH 7.4 PBS/S/T were added, and incubated for 1 h at RT. The plates were washed four times as above, before alkaline phosphatase-conjugated polyclonal anti-HSA antibody from goat (#A80229AP; Bethyl Laboratories, Inc., 1:5000), diluted (1:3000) in pH 5.5 or pH 7.4 PBS/S/T was added for 1 h at RT. The plates were washed four times and developed as above.

**MicroScale Thermophoresis assay.** A Monolith NT.115 instrument (Nano Temper Technologies GmbH, Munich, Germany)[67] was used where anti-NIP hIgG1 variants labelled with NT-647-HNS fluorescent dye (Nano Temper Technologies) or C-terminally EGFP-fused HSA variants were used. Titrated amounts of hFcRn (60,000–0.3 nM) were incubated with 20 nM of each of the labelled hIgG1 variants or 100 nM HSA-EGFP at RT for 10 min in PBS pH 7.4 or 100 mM phosphate buffer (6 mM $Na_2HPO_4 \times 2H_2O$, 94 mM $NaH_2PO_4 \times H_2O$, 150 mM NaCl) pH 5.5, both buffers supplemented with 0.01% T. Samples were loaded onto premium-coated capillaries (Nano Temper Technologies). Measurements were performed at 25 °C using 40% LED and 40% MicroScale Thermophoresis power for hIgG1 variants, and 80% LED and 60% power for HSA-EGFP variants. All experiments were repeated three times for each measurement, and data analyses were done using Nano Temper analysis software.

**Analytical hFcRn affinity chromatography.** Analytical hFcRn affinity chromatography was performed using an ÄKTA Avant 25 instrument (GE Healthcare), essentially as previously described[58,68]. Briefly, 50 μl WT hIgG1 and Fc-engineered variants (1 mg/ml) were injected and eluted by a linear pH gradient from pH 6.0 to 8.8 within 110 min using 20 mM MES sodium salt, 140 mM NaCl 5.5 and 20 mM Tris/HCl, 140 NaCl, pH 8.8 as eluents. To determine the elution pH at particular retention times, the pH was monitored by a pH detector (GE Healthcare). HSA variants were analysed using 2 mg/ml followed by a pH elution gradient from pH 5.5 to 8.8 within 110 min using the same buffers as above.

**siRNA knockdown of FcRn expression.** HMEC1-hFcRn cells were transfected with a mixture of three control siRNAs or hFcRn HC-specific siRNAs (sc-37007, sc-45632; Santa Cruz Biotechnology Inc). The sequences of hFcRn HC-specific siRNAs are shown in Supplementary Table 2. For each transfection, a siRNA mixture was diluted in siRNA transfection medium and transfection reagent, essentially as described by the manufacturer (Santa Cruz Biotechnology Inc). Cells were incubated for 6 h at 37 °C in a $CO_2$ incubator followed by adding MCDB 131 medium with 20% FCS and 200 U/ml PS. Cells were then incubated for additional 24 h before medium was replaced with MCDB 131 medium.

**Confocal laser scanning microscopy.** HMEC1-hFcRn-EGFP cells were seeded in eight-well NuncTM Lab-Tek™ chambered Coverglass imaging dishes (Thermo-Fisher Scientific), and grown to 50–70% confluency. WT and IHH hIgG1 variants were conjugated with Alexa-647 following the manufacturer's procedure (Life Technologies). Cells were washed three times with HBSS pH 6.0 and Alexa-647 conjugated antibodies, diluted in HBSS pH 6.0 to the final concentration of 400 nM, were added to the cells and incubated for 4 h before pictures were taken. The cells were then washed three times with HBSS pH 7.4, and incubated in HBSS pH 7.4 for 4 h before pictures were taken. Lysotracker DND-99 (Life Technologies) was added to the cells 30 min before pictures were taken. Confocal images were acquired on an Olympus FluoView 1000 inverted microscope equipped with a PlanApo 60/1.42 oil objective (Olympus). Constant temperature was set to 37 °C and $CO_2$ to 5% by an incubator enclosing the microscope stage. Image acquisition was done by sequential line scanning to eliminate bleed-through. Images were processed and prepared with ImageJ (NIH), Adobe Photoshop and Illustrator

# ARTICLE

(Adobe system Inc). Co-localization was quantified using Imaris spot co-localization software (Bitplane) where the endosome size was set to 1 μm. Data from two independent experiments with 15–20 cells analysed were used for co-localization analysis.

**In vivo studies.** Hemizygous Tg32 mice (B6.Cg-*Fcgrt*tm1Dcr Tg(*FCGRT*)32Dcr/DcrJ; The Jackson Laboratory) that are transgenic for the hFcRn gene and WT mice (BALB/c; Taconic) were used to determine the half-life of the hIgG1 antibodies, except for YTE/KF that was evaluated in homozygous Tg32 mice. Male mice, age 7–9 weeks, weighing between 17 and 27 g (5 mice per group), received 5 mg/kg of hIgG1 variants diluted in PBS by intravenous injections. Blood samples (25 μl) were drawn from the retro-orbital sinus of the Tg32 mice at days 1, 3, 5, 7, 10, 12, 16, 19, 23, 30 and 37 after injection. WT mice had blood drawn from the saphenous vein on days 0, 1, 2, 4, 6, 8, 10 and 12 after injection. The blood samples were immediately mixed with 1 μl 1% K3-EDTA to prevent coagulation and then centrifuged at 17,000 × *g* for 5 min at 4 °C. Plasma was isolated and diluted 1:10 in 50% glycerol/PBS solution and then stored at −20 °C until analysis by ELISA as described above. Plasma samples were diluted 1:400 in PBS/S/T and 100 μl was added per well in the ELISA plate. The Tg32 mice studies were carried out at The Jackson Laboratory (JAX Services, Bar Harbor, ME), while the WT mouse study was done at the Department of Immunology, Oslo University Hospital, Rikshospitalet. The experiments and procedures used were all approved by the Animal Care and Use Committee at The Jackson Laboratory (Tg32 study) and the Norwegian Animal Research Authority (BALB/c study), and were performed in accordance with the approved guidelines and regulations.

**Half-life calculation.** The plasma concentration of the hIgG1 antibodies was presented as percentage remaining in the circulation at different time points after injection compared to the concentration on day 1 (100%). Nonlinear regression analysis was performed to fit a straight line through the data using Prism 7. The $\beta$-phase half-life was calculated using the formula: $t_{1/2} = \log 0.5/(\log A_e/A_0) \times t$, where $t_{1/2}$ is the half-life of the hIgG1 evaluated, $A_e$ is the amount of hIgG1 remaining, $A_0$ is the amount of hIgG1 on day 1 and $t$ is the elapsed time.

**Statistical analysis.** Figures were generated and statistical analyses (ANOVA with Tukey's multiple comparisons test or Dunnett's multiple comparison test, with a 95% confidence interval and $p < 0.05$ was considered a statistically significant difference) were performed using GraphPad Prism 7 for Windows (Version 7.02; GraphPad Software Inc.) and Microsoft Excel 2010 (Microsoft).

**Data availability.** The data that support the findings of this study are available from the corresponding author upon reasonable request.

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

## Acknowledgements

We are grateful to Sathiaruby Sivaganesh for excellent technical assistance. This work was supported in part by the Research Council of Norway through its Centre of Excellence funding scheme (Project 179573). J.T.A. and J.N. were supported by the Research Council of Norway (Grants 230526/F20 and 179573/V40). M.B.M. was supported by the Research Council of Norway (Grant 179573/V40). M.B. was supported by the Research Council of Norway through its program for Global Health and Vaccination Research (Grant 143822). S.F. and A.G. were supported by the University of Oslo. We thank Dr. Wayne I. Lencer (Boston Children's Hospital, Harvard Medical School and Harvard Digestive Diseases Center, USA) for the WT HMEC1 and the HMEC1 cell line stably expressing HA-hFcRn-EGFP and Dr. Finn Eirik Johansen (University of Oslo, Norway) for the J558L cell line stably expressing NIP-specific hIgG1-IHH. We acknowledge the use of the NorMIC Oslo imaging platform at the Department of Biosciences, University of Oslo.

## Author contributions

A.G., M.B.D., I.S. and J.T.A. designed research; A.G., K.M.K.S, J.N., G.J.C, M.B., M.B.D., S.F., M.B.M. and I.Ø. performed research; A.G., K.M.K.S, J.N., M.B.D., S.F., T.S., T.E.M., R.S.B., D.C.R. and I.Ø. analysed data; A.G., I.S. and J.T.A. wrote the paper.

## Additional information

**Competing interests:** The authors declare no conflicting financial interests.

