## [Peer Review File · Nature Communications]

Reviewers' comments:

Reviewer #1 (Remarks to the Author):

The authors describe an interesting and valuable cellular assay to investigate the mechanisms which determine serum half-lives of proteins. Specifically, they use different FcRn mutants with altered affinities towards their natural ligands, namely IgG antibodies and albumin, and show that pH-dependent changes in affinity correlate nicely with their cellular assay.

The data from the cellular assay are backed by biochemical and biophysical analysis, namely analytical affinity chromatography and MST, both of which correlate with the cellular results. This paper therefore not only provides an interesting approach to screen engineered antibodies or HSA for half-lives, but also adds information about the mechanisms responsible for different serum half-lives in vivo.

I therefore recommend publication in Nature Communications

Reviewer #2 (Remarks to the Author):

The manuscript describes the application of a human endothelial cell based assay system proposed as a screening tool to identify IgG or albumin-fusion constructs with 'better' pharmacokinetic properties based on FcRn interaction.

Overall, the authors did a thorough job of demonstrating that the assay system is reflective of FcRn mediated recycling/rescue using both IgG and albumin. This was accomplished by studying the system with molecules employing mutations in the IgG-Fc and albumin known to influence IgG-FcRn interaction as well as siRNA knock down. The cellular measurements were defined and results related reasonably with in vivo pharmacokinetic parameters determined in transgenic mice.

There are several questions and comments that I'd like the authors to address in the manuscript before being considered for final publication.

Comments:

The value of the system seems to be in its reflection of multiple intracellular processes and mechanistic insights which may be more representative of less 'biologically' relevant measures.

I would suggest that consideration be given in the discussion to the following points.

The liability of residual/increased binding or lack of dissociation at pH 7 as a negative factor from a PK point of view is pretty well established in the literature. It's also becoming pretty well established that charge related interactions, not related to FcRn, override potential benefits of improved FcRn binding. We and others have also shown that general non-specific binding assays (plate bound or cellular) can identify molecule attributes to be avoided to attain reasonable in vivo PK performance. From the perspective of identifying IgGs with appropriate PK properties it's not obvious why this offers significant advantage in selection vs. analytical approaches to characterize biophysical properties or FcRn interactions (such as the FcRn column). It would be useful if the authors put the benefits and/or additional insights of the HERA assay into context with these approaches.

I would agree that the system offers opportunity to characterize mechanisms. For albumin or Fc-fusions, aspects how intracellular trafficking is influenced may offer insights on how to optimize a molecule. A key benefit could be guiding SAR by defining potential degradation of the 'active' peptide/protein intracellularly. I'd suggest altering the discussion to focus on utility for more 'mechanistic' evaluations vs. screening.

Questions:

1. What is the relative level of FcRn expression in the stably transfected cells vs. endogenous FcRn expression? Would this potentially influence how one could differentiate constructs?
2. How were concentrations/time points decided upon? The imaging expts and cellular recycling expts are markedly different, is there potential influence to the interpretations?
3. I don't understand the interpretation of results from Figure 3c and 3d (lines 151-155). If I read this correctly, IgG-IHH signal could not be detected during uptake, but could be detected after the buffer exchange.
4. In the cell-assay results much less than 0.5% of added IgG (WT or variants) is taken up by the cells even with more directly facilitated binding at pH 6. Outside of YTE/KF, most of the data for the other variants are similar.
Given this, it isn't compelling to make the argument for this being a robust or sensitive analysis if for comparing/screening native IgGs without up mutations, since differences here are likely to be subtle.
5. In Figure 5, mass balance is generally not obtained. That is, residual + recycled doesn't equate to amount of uptake. Is the difference a reflection of degradation? It'd be nice to put that into context with the binding interactions determined with the variants.
6. In discussion of Figure 6, it'd be useful to describe the binding kinetics of these molecules on mFcRn to clarify the reasons for discrepancy in WT mice.
7. While I agree that normalizing an assay would be useful in regard to the comment about the FDA requirement for FcRn binding kinetics. In general however, at this stage the data is descriptive and generally doesn't have context in isolation. To me this approach would not really provide any additional benefit from a reporting perspective.

Reviewer #3 (Remarks to the Author):

The manuscript entitled 'A human endothelial cell-based recycling assay (HERA) for rapid screening of FcRn-mediated rescue from degradation of IgG and albumin' Grevys et al presents a novel in vitro method to study the interaction of IgG1 and albumin with hFcRn on the cellular level. The study demonstrates that a panel of antibodies modified to increase FcRn binding behaves the same in the novel cellular assay as in a hFcRn transgenic mouse PK study. This makes this new method an interesting and straightforward assay to predict FcRn-mediated half-life extension and for the study of Abdegs,
Finally, it is claimed that albumin and antibody binding to FcRn can take place at the same time and do not interfere with each other's binding to FcRn.
However, some issues temper my enthusiasm for this study:

Major points:

- 1) In the current set-up, the 2 studied proteins are naturally interacting with FcRn and modified variants thereof. As is indicated in the discussion, the field would however benefit from proof that the presented system also works with proteins to which FcRn binding was recombinantly added. Furthermore, the field would benefit from a model to study FcRn-mediated transcytosis. I believe that this model could also show those properties in, for example, a transwell set-up.
- 2) To show the importance of FcRn in the HERA system for recycling, 3 different siRNA's are used. Could the researchers also show that there indeed was a concomitant decrease in FcRn expression on the protein level?
- 3) The HERA system uses the HMEC-1 cell line. Since the HMEC-1 cells already express FcRn intrinsically, I wonder whether the expression of FcRn is increased in comparison to WT cells after stable transfection with HA-hFcRn-EGFP? Is there competition between fluorescent FcRn and endogenously present FcRn?

4) In figure 2a, there is an abundance of fluorescent signal. The overlay shows overlap of signals and therefore it is claimed that co-localization takes place. It would add to the quality of the paper to use FRET in this setting to show true co-localization between FcRn and IgG.

5) In figure 4, microscale thermophoresis is used to determine the affinity between FcRn and different antibodies. Would a cellular based assay where FcRn is in its natural conformation and is present together with B2m be a better method of analyzing affinity?

What is the reason the studied pH now significantly differs in comparison to the previous experiments (5.5 instead of 6)? Would incubating at pH 5.5 make the assay more sensitive?

6) An uptake period of 4 hours is chosen, in which recycling already will take place. It would add to the paper to show more details of the kinetics of uptake and recycling.

7) In figure 8, a saturating amount of antibody is added to show that albumin is still able to be properly recycled by FcRn. It would add to the quality of the paper to show that in a context of saturation with HSA, recycling of antibodies is also taking place.

Other remarks:

- Can the authors give explanation why in figure 3 another timeframe is chosen in comparison to the other assays for uptake and recycling?

- Does the HERA score of HSA and K573P HSA correlate also with the half-life? This would be worth adding to figure 7

- In the text there is being referred to figure 5C: "to the relative amounts of antibodies remaining inside the cells after additional 4h incubation at pH 7". It would seem that this should be figure 5F.

Comments on figures:

- Figure legend of figure 1d: What do the researchers mean with "before 0 h"? This seems to be an error and should be corrected.

- Figure 2: Clarity would improve when "concentration antibody" would be added to the x-axis of figure a and b. Also add that figure D-I are at pH 7.4

- Figure 3: It is difficult to understand that the added picture legends indicate green or red signals for FcRn/Antibody, but are displayed in white, but that these colors are shown in the third panel.

Furthermore, in the middle panel of figure 3d an insert is still visible without any function.

- Figure 4: Why is the analytical hFcRn affinity chromatography profile called a "SEC profile"? It seems that this is an error and should be corrected.

- Figure 5: Why are there a lot less data points in figure D-F if these points are derived from A-C? In addition, a part of the bar in figure 5F is missing. Finally, the data in figure 5b and figure 5g seem to show groups for the same antibody behaving very differently. Please clarify.

- Figure legend figure 6: I think before you mention "5 animals", some text was deleted. This should be corrected.

Statistics:

Figure 5: Why is statistical analysis only applied for pH 7.4?

Figure 8: Please draw a line for significance between WT and YTE/KF.

Spelling errors:

Although the text is clearly written most of the time, a number of spelling/grammatical errors appear in the text.

Methodology:

The provided detailed methodology allows that the experiments can be reproduced.

Conciseness:

The paper shows the necessary figures to state its point.

Reviewer #4 (Remarks to the Author):

The manuscript (MS) of Jan Terje Andersen (corresponding author) entitled as "A human endothelial cell-based recycling assay (HERA) for rapid screening of FcRn-mediated rescue from degradation of IgG and albumin" reports on experiments that can facilitate the analysis of the half-life of monoclonal antibody (IgG) and albumin (or its derivatives). The manuscript is well designed, contains important information to FcRn specialists, and to scientists in other related disciplines. Monoclonal antibodies (IgGs) are important therapeutics and one of their important features is their long half-life in the body due to interaction of their Fc-domain (CH2-CH3) with the neonatal Fc receptor (FcRn) that protects them from fast degradation, primarily in the capillary endothelial cells. Because of the long half-life, there are several Fc-fusion therapeutics in the clinic or in developmental phases. Similarly, albumin binds to this receptor and is also protected from fast degradation and thus has similarly long half-life. Therefore, there are major interest to use albumin or part of it, as albumin-conjugated therapeutics.

IgG and albumin protection is a stunning biological process. It is widely accepted, that all kind of molecules from the serum are uptaken by capillary endothelial cells (and the manuscript focuses only to capillary endothelial cells) through non-specific pinocytosis and then reach their early endosomes. Endosome maturation leads acidification and that results in binding of IgG and albumin to FcRn. Importantly, recombinant molecules in biosensors revealed that IgG and albumin bind two different regions of FcRn and therefore, there is no competition or influence in binding between these two ligands. These interactions are followed by recycling, i.e. those vesicles containing IgG- and/or albumin-FcRn complexes are transported to the cell surface where these vesicles temporarily merge to cell membrane and the physiological pH of the blood dissociates IgG and albumin from the FcRn. Consequently, these two macromolecules are released from the endothelial cells and got back to bloodstream. Those uptaken molecules that do not bind to FcRn degrade in lysosomes. Whether bindings of IgG and albumin to FcRn occur in parallel in nature and how do they affect the FcRn function in recycling these two molecules has not been tested in cells, previously. Also, there are some studies indicating that IgG uptake is not FcRn independent and thus these questions are important aspects in FcRn research and from the point of therapeutic IgG, albumin or their derivatives development.

There are several studies indicating how important is this pH dependent binding and dissociation in IgG and albumin half-life. An IgG that cannot bind to FcRn at slightly acidic pH (pH ~6) has very short half-life. Also, some mutated IgGs that have improved binding at pH 6 have longer half-life. On the other hand, it is important to have minimal or no affinity of IgG or albumin to FcRn at physiological pH to elongate half-life. An IgG that binds to FcRn at pH ~7.4, remain trapped with FcRn and thus their half-life will be short (and even block FcRn to protect other IgGs). Alike to IgG, albumin has very similar phenotype. In the field of FcRn research, this important characteristic is referred as "IgG and albumin uptake and release as a function of pH".

Current methods that analyze pH dependent bindings of these molecules are biosensors (e.g. SPR) and affinity chromatography with pH elution gradient from slightly acidic to slightly alkaline pH. It is well known, that affinity data derived from biosensor studies are not always predict in vivo half-lives of different IgG variants. Importantly, there is no available in vitro cell system that would functionally and quantitatively characterize the interaction of IgG/albumin to FcRn, regarding their

interaction dependence at both pHs.

I. The current report has the following novel and important findings:

- 1) HERA screening provides a new in vitro tool to characterize the IgG and albumin uptake and release as a function of both pH and the score that is presented here is in line with in vivo half-life studies in hFcRn transgenic mice. The advantage of the presented experimental design is that differently mutated Fc regions of IgGs were used, as the affinity of the binding of the mutants gradually increase at both pH 6 and 8 (wild type and 4 different mutants). In case of albumin, the wild type and one mutant was used with increased affinity at both pHs.
- 2) The in vitro cell-based studies show that IgG uptake depends on binding affinity of IgG-FcRn, as IgG with increased affinity results in increased uptake at both pH 7.4 and 6 (Figure 5). Also, an IgG mutant that doesn't bind to FcRn cannot be uptaken in this cell model (Figure 3). These data highlight an ongoing dispute in the field (whether FcRn is located in the cell membrane or not) and thus demands further experiments before accepting this manuscript.
- 3) Eloquent in vitro cell studies prove that IgG and albumin are recycled/transported by FcRn in parallel and they do not influence the transport of each other.

II. Major concerns:

- 1) This is a complex manuscript with several well-structured experiments. Nevertheless, reviewer believes that the manuscript could be improved by emphasizing the unknown issues of the function of FcRn that are targeted and investigated. Neither the abstract nor the introduction are focused enough and thus should be rewritten.
- 2) It seems that the IgG mutant that does not bind to FcRn (IgG IHH) are not uptaken by HMEC1-hFcRn cells (Figure 3). This observation is mentioned, somewhat discussed (page 6, line 152-153; and page 11, line 290-291) but not resolved. In parallel, IgG mutants that have higher affinity to FcRn have higher uptake. These data can be explained only if FcRn is involved in IgG uptake (and indicates that IgG uptake is not simply based on non-specific pinocytosis). Also, reviewer disagrees with the argument that the non-visible signal of the uptake of this mutant might be the result of the fast degradation of IgG, as Alexa647 does not fade in the lysosome (as this stain is pH insensitive) and thus could have been visible even if IgG is degraded. If there is no sign of Alexa647 in these cells that means no uptake of Alexa647-labeled IgG. Reviewer requests integrating additional experiments to clarify this important question. Perhaps shorter uptake (eg. 15 min, 30 min, 1 hr, 2 hr at both pH) with appropriate controls should be applied, or using FcRn downregulated cells.

Minor concerns:

- 1) Figure 2. legend should indicate the function of the different monoclonal antibodies (DVN24, ADM31, ADM12) – although manuscript provides necessary information in this regards, this info in figure legend would help readers to understand this figure without reading the related text.
- 2) Figure 5. scales are different and thus confusing a bit (especially inlet a versus g, or c versus i); also, reviewer would like to see data of uptake of IgG-IHH (that does not bind to FcRn).
- 3) Figure 8. it seems that descriptions of inlet "a" and "b" are switched: reviewer thinks that "a" indicates relative uptake while "b" refers to relative recycling" (as in the Y axis of the referred figures)
- 4) Table 2 presents serum half-life data in wt and humanized FcRn mice. Please explain why IgG mutants have been analyzed in two different mice, i.e. Tg32 hemizygous (only one allele express human FcRn) and Tg32 homozygous (both alleles express human FcRn). Theoretically, IgG half-life depends on many factor, including the level of FcRn expression.

In conclusion, reviewer supports the publication of this manuscript and suggest some additional experiments and amendments of the text.

Point by point responses to reviewers

We have performed additional experiments and analysis as requested. The authors thank the reviewers for their helpful comments, which provided us with critical input that enabled us to improve the manuscript. A full point-by-point response to the reviewers' comments can be found at the end of this letter. Please note that the modifications in the manuscript are highlighted in yellow.

Reviewer #2:

1. *The liability of residual/increased binding or lack of dissociation at pH 7 as a negative factor from a PK point of view is pretty well established in the literature. It's also becoming pretty well established that charge related interactions, not related to FcRn, override potential benefits of improved FcRn binding. We and others have also shown that general non-specific binding assays (plate bound or cellular) can identify molecule attributes to be avoided to attain reasonable in vivo PK performance. From the perspective of identifying IgGs with appropriate PK properties it's not obvious why this offers significant advantage in selection vs. analytical approaches to characterize biophysical properties or FcRn interactions (such as the FcRn column). It would be useful if the authors put the benefits and/or additional insights of the HERA assay into context with these approaches.*

We thank the reviewer for the comment. We agree that although HERA represents a novel cellular assay for screening of FcRn-mediated transport and rescue from degradation of IgG and albumin variants and derived scaffolds, it will not always answer all possible questions related to prediction of *in vivo* half-life of monoclonal antibodies. However, in general, we strongly believe that HERA is an attractive novel tool that can be used to compare antibodies with the same framework/specificity, and that have structural alterations such as Fc amino acid substitutions.

As the reviewer points out, recent reports have demonstrated that poor PK may be due to FcRn-independent factors such as "promiscuous" binding or cross-reactivity. However, other reports claim that such drawbacks may be due to altered FcRn binding kinetics. Regarding off-target effects due to cross-reactivity, antibodies may be screened toward panels of non-cognate proteins. Such screens can be done prior to HERA, or may be combined with HERA, to gain insights into how structural differences are affecting cellular uptake and FcRn-mediated recycling. We have elaborated on this important topic in the Discussion (page 14).

To further highlight that HERA is a tool that can be used to gain insights into how antibodies with different structural features are behaving, we have in an on-going study addressed how charge differences in the Fab arms, and in particular in the CDR loops, are affecting cellular uptake and FcRn-mediated transport. Interestingly, for some of the antibodies with unusual short *in vivo* half-life, prolonged half-life can be restored by introducing Fc substitutions that alter FcRn binding kinetics. HERA gives important insights into how such antibodies behave in a cellular system. As these exciting preliminary data are part of a larger project, we hope that the reviewer agrees that they are beyond the main scope of the current manuscript.

2. *I would agree that the system offers opportunity to characterize mechanisms. For albumin or Fc-fusions, aspects how intracellular trafficking is influenced may offer insights on how to optimize a molecule. A key benefit could be guiding SAR by defining potential*

degradation of the 'active' peptide/protein intracellularly. I'd suggest altering the discussion to focus on utility for more 'mechanistic' evaluations vs. screening.

We fully agree that HERA is not only a great tool to screen IgG and albumin variants, but also can be combined with other methods to gain a more in-depth understanding of the mechanisms of intracellular trafficking events. A benefit of HERA is that the receptor is expressed with a C-terminal EGFP-tag that can be used for imaging studies. Regarding imaging of FcRn-mediated transport, Sally Ward and her colleagues have contributed with elegant and advanced imaging studies during the last 10-15 years on how FcRn is transporting IgG variants with different binding properties. In our case, we focus on HERA combined with imaging and other techniques to understand how FcRn is transporting its two ligands and therapeutics built on these molecules. The possibility for further mechanistic studies by combining HERA with imaging has been made clear in the Discussion (page 12).

3. What is the relative level of FcRn expression in the stably transfected cells vs. endogenous FcRn expression? Would this potentially influence how one could differentiate constructs?

This is a highly relevant question. HERA is a sensitive and robust assay where ligands are given in low amounts without labeling. The ligands can then be detected inside the cells and in the medium following recycling by the use of simple ELISAs run on collected medium and cell lysate fractions. For this to be possible, it requires that human FcRn is over-expressed. We have determined the expression level of FcRn in the transfected HMEC1 cells and compared that with the level of expression in the parental cell line. This was done by extracting the membrane protein fraction from the cell lines, before adding it to an ELISA, where functional receptor was captured on hIgG1-YTE/KF coated in wells, followed by detection using a monoclonal anti-human FcRn antibody (ADM31). Titrated amounts of soluble recombinant human FcRn were included as a standard to quantify the levels of functional receptor in the isolated fractions. The results showed that the transfected cell line expresses about 100-fold more FcRn than the parental cell line. Using the established HERA protocol, recycling is not possible to measure using the parental cell line. The data on quantification of the expression levels have been added to Results (page 5) and Supplementary Fig. 1a-c.

4. How were concentrations/time points decided upon? The imaging expts and cellular recycling expts are markedly different, is there potential influence to the interpretations?

We added 400 nM IgG to cells, incubated at pH 7.4, and detected uptake after 1 h, 2 h, 3 h or 4 h. Cell lysates were collected and added to ELISA. Based on this, 4 h was chosen. Then, the same experiment was performed to measure the amounts in medium over time, which is the result of recycling, after adding 400 nM of antibody followed by incubation for 4 h, extensive washing and additional 1 h, 2 h, 3 h or 4 h incubation time before medium samples were collected. Based on this experiment, we chose 4 hours for the second step as well. The data have been added to Results (page 5) and Supplementary Fig. 2a-b. As highlighted in Results, this step may also be run overnight without affecting the relative differences of the antibodies tested (page 7, Supplementary Fig. 5a-c).

We agree with the referee that it is unfortunate that the imaging experiments were performed using a set-up that was not the same as that used for HERA. We have repeated the imaging study with the same timeline as for HERA, and the conclusions remain the same. We did also include a lysosomal tracker (LysoTracker) to visualize the antibody fractions localized in to

lysosomes. In addition, we have quantified co-localization of the antibodies with the receptor positive endosomes and with lysosomes. The imaging figures (Figure 3a-h and Supplementary Fig. 4a-d) have been replaced and Results (page 6) updated in accordance with the findings.

5. *I don't understand the interpretation of results from Figure 3c and 3d (lines 151-155). If I read this correctly, IgG-IHH signal could not be detected during uptake, but could be detected after the buffer exchange.*

The imaging experiments have been repeated using the same time points as in HERA, where we included lysotracker to visualize localization of the antibodies in lysosomal compartments. In addition, co-localization of hIgG1 and FcRn positive endosomes or lysosomes were quantified using Imaris spot co-localization analysis (Fig. 3a-h; supplementary Fig 3a-d). The new results confirm the previous conclusions. Figure 3 has been replaced and the result updated.

The data show that roughly 3 times more endosomes are positive for WT hIgG1 than the IHH variant when uptake was detected after 4 hours at pH 6. At this pH, WT hIgG1 can engage FcRn at or near the cell membrane, and the receptor will be actively involved in uptake. This will lead to more efficient uptake/accumulation of IgG WT at pH 6 than at pH 7.4. In contrast, IHH will not be actively internalized via FcRn at pH 6, since it does not bind the receptor. Thus, uptake is markedly different between the antibody variants at pH 6 (Fig. 3a-d). When uptake is done at pH 7.4, WT hIgG1 will be taken up in an FcRn-independent manner. Thus, at pH 7.4, both WT IgG and IHH will be taken up by pinocytosis, and the amounts of uptake are more similar between the two variants (Supplementary Fig. 4a-d) compared to the experiments done at pH 6 (Fig. 3a-h).

6. *In the cell-assay results much less than 0.5% of added IgG (WT or variants) is taken up by the cells even with more directly facilitated binding at pH 6. Outside of YTE/KF, most of the data for the other variants are similar.*

7. *Given this, it isn't compelling to make the argument for this being a robust or sensitive analysis if for comparing/screening native IgGs without up mutations, since differences here are likely to be subtle.*

The assay was established to be sensitive and robust and not require labeling of the ligands. In addition, simple ELISA set-ups should be used as readout. Therefore, we established a protocol where small amounts of the WT ligands are added, so as to achieve reasonable levels inside the cells at a given time. Specifically, the amounts required were determined to be 400-800 nM followed by an incubation step of 4 hours before extensive washing. The amounts of IgG/albumin present inside the cells at this time are defined as "uptake". Our results demonstrate that the amounts present inside the cells as well as in the medium at a given time differed for the WT and Fc-engineered IgG1 variants with the same specificity and scaffold, and that only differed by a few amino acid substitutions in their Fc region. Thus, their behavior in HERA is a direct result of the Fc substitutions. We respectfully disagree with the statement that only YTE/KF shows distinct behavior and not the other variants, as we demonstrate that the IgG1 variants behave differently, which is in line with altered FcRn binding kinetics and *in vivo* half-life values obtained from human FcRn Tg mice.

8. *In Figure 5, mass balance is generally not obtained. That is, residual + recycled doesn't equate to amount of uptake. Is the difference a reflection of degradation? It'd be nice to put that into context with the binding interactions determined with the variants.*

We thank the reviewer for raising this question. It is correct that residual + recycled doesn't equate to amount of uptake, which may indeed be due to degradation. We have clarified this in Discussion (page 11).

9. *In discussion of Figure 6, it'd be useful to describe the binding kinetics of these molecules on mFcRn to clarify the reasons for discrepancy in WT mice.*

Despite the similarities in amino acid composition of the mouse and human FcRn heavy chains and conservation of key residues on the IgG Fc across species, the binding affinities specificity of human and mouse FcRn for their ligands are very distinct. For instance, mouse FcRn binds promiscuously to IgGs from multiple species including human IgG1, whereas human FcRn is much more selective. Importantly, the binding affinity of mouse FcRn for human IgG1 is about 10-fold higher than that of the corresponding human FcRn. This is of relevance when considering testing human IgGs in mouse models. Although the mouse FcRn-human IgG1-WT interaction retains pH dependence such that long half-life is achieved in mice, this is not the case for human IgG Fc-engineered variants with higher affinity for FcRn. For example, a human IgG mutant can acquire significant binding to mouse FcRn at near neutral pH, while retaining the necessary low affinity for human FcRn to allow efficient recycling in human systems. In conventional mice, such IgGs may have shortened *in vivo* half-lives and inhibit FcRn function, while having longer half-life in human FcRn transgenic mice, non-human primates and humans. Thus, conventional mice are not suitable models for half-life evaluation of engineered human IgG variants. We have made this important issue clear in the Introduction (page 3-4) and Discussion (page 11-12). In addition, we have included ELISA results that pin-point the differences in pH dependent binding between the mouse and human forms of FcRn toward the engineered human IgG1 variants included in this study. The data has been added to Results (page 8-9) and Supplementary Fig. 7a-d.

10. *While I agree that normalizing an assay would be useful in regard to the comment about the FDA requirement for FcRn binding kinetics. In general however, at this stage the data is descriptive and generally doesn't have context in isolation. To me this approach would not really provide any additional benefit from a reporting perspective.*

HERA may be one possible assay to solve and overcome this challenge, but at this stage we agree that it may be too early to conclude that this is the final answer to the requirement from FDA. We have deleted the statement in Discussion (page 14).

Reviewer #3:

1. *In the current set-up, the 2 studied proteins are naturally interacting with FcRn and modified variants thereof. As is indicated in the discussion, the field would however benefit from proof that the presented system also works with proteins to which FcRn binding was recombinantly added. Furthermore, the field would benefit from a model to study FcRn-mediated transcytosis. I believe that this model could also show those properties in, for example, a transwell set-up.*

We agree with the reviewer that it would be of interest to use HERA to screen different IgG and Fc formats with or without fusions as well as albumin fusions and conjugates. This is indeed an on-going project in our lab, where we have made panels of fusions and conjugates combined with engineering for altered FcRn binding. Preliminary data reveal large differences in their ability to be transported and rescued from degradation. Data on "lead"

candidates will be extended to and compared with half-life data obtained in state-of-the-art human FcRn transgenic mice. In addition, we are addressing how the presence of an antigen of difference size is affecting transport. As these studies are far from finished and include engineered variants that we at this time cannot disclose due to issues related to intellectual properties as well as collaboration with industrial partners, we hope that the editor and reviewer agree that such a large set of experiments are beyond the scope of the current manuscript.

We agree that it may also be of interest to test how IgG Fc-engineered variants are transported across cellular layers. Recently, we reported on such a Transwell assay based on the human polarizing epithelial cell line T84, which can be used to measure bi-directional transcytosis of both IgG1 and Fc-fusions (J Control Release. 2016 Feb 10;223:42-52). Here, we found that the efficacy of transport was dependent on the format and that transepithelial transport could be enhanced by Fc-engineering for improved FcRn binding as well as by Fc-polymerization. Efficient transcellular delivery across human epithelial cells was shown to require optimal pH-dependent FcRn binding that can be manipulated by avidity and Fc-engineering. Like HERA, this assay is also independent of labeling of the ligands and ELISA is used as readout. That said, the Transwell system is much more time consuming to operate and very expensive to use for screening compared with the simple and easy HERA protocol. We are in the process of comparing how a panel of selected IgG variants is behaving in the Transwell system compared with HERA. This is of particular interest to understand how stoichiometry and avidity are affecting transcytosis versus recycling. Recently, colleagues at Roche and UCB Celltech also reported on an MDCK cell based Transwell system that could be used to compare transport of IgG variants. However, in this case, the IgG variants were 3H-labeled prior to testing (MAbs. 2017 Jul;9(5):781-791).

2. To show the importance of FcRn in the HERA system for recycling, 3 different siRNA's are used. Could the researchers also show that there indeed was a concomitant decrease in FcRn expression on the protein level?

We thank the reviewer for this comment. We have added data showing downregulation of FcRn at the protein level following siRNA treatment (Supplementary Fig. 3a-b). The level of expression of FcRn after siRNA treatment was probed by both Western blotting and ELISA. Briefly, the membrane protein fraction was isolated from the HMEC1-hFcRn cells after 24h, 48h, 72h, 96h or 120 h treatment with the siRNA mixtures. Then, the amounts of FcRn were quantified by capturing functional receptor on hIgG1-YTE/KF coated in wells followed by detection using a monoclonal anti-human FcRn antibody (ADM31). Titrated amounts of soluble recombinant human FcRn were included as a standard to quantify the levels of functional receptor in the isolated fractions. In Western blotting, the over-expressed human FcRn was detected using polyclonal anti HA-tag antibody (The heavy chain contains an N-terminal HA-tag). The data show that siRNA treatment to knockdown FcRn is very efficient and last for 3-4 days. Results (page 5) have been updated and data added to Supplementary Fig. 3a-b.

3. The HERA system uses the HMEC-1 cell line. Since the HMEC-1 cells already express FcRn intrinsically, I wonder whether the expression of FcRn is increased in comparison to WT cells after stable transfection with HA-hFcRn-EGFP? Is there competition between fluorescent FcRn and endogenously present FcRn?

This is an important question that was also raised by reviewer #2 (3): HERA is a sensitive and robust assay where ligands can be in given in low amounts without labeling. The ligands

can then be detected inside the cells and in the medium following recycling by the use of simple two-way ELISAs. For this to be possible, it requires that human FcRn is over-expressed. We have determined the expression level of FcRn in the transfected HMEC-1 cells and compare that with the level of expression in the parental cell line. This was done by extracting the membrane fractions from the cell lines before added to an ELISA set-up where functional receptor was captured on hIgG1-YTE/KF coated in wells followed by detection using a monoclonal anti-human FcRn antibody (ADM31). Titrated amounts of soluble recombinant human FcRn were included as a standard to quantify the levels of functional receptor in the isolated fractions. The results showed that the transfected cell line expressed about 100-fold more FcRn than the parental cell line. Using the established HERA protocol, recycling is not possible to measure using the parental cell line. The data on quantification of the expression levels have been added to Results and Supplementary Fig. 1a-c.

4. In figure 2a, there is an abundance of fluorescent signal. The overlay show overlap of signals and therefore it is claimed that co-localization takes place. It would add to the quality of the paper to use FRET in this setting to show true co-localization between FcRn and IgG.

FRET can be used to monitor complex formation between two molecules where one is labeled with a donor and the other with an acceptor. This will then require a pair of fluorophores added to the ligand and to the receptor, and both fluorescent probes have to be in close proximity. In our system it would not be optimal to use the established HMEC-1 cell line, since fluorescent probes will be separated by the endosomal membrane, as EGFP is added to the C-terminal end of the FcRn heavy chain. Thus, to use FRET, we would need to make a new vector where for instance the cDNA encoding EGFP is added in frame with the sequence encoding the N-terminal of the FcRn heavy chain. How this will affect binding to the ligands is unknown. Although we agree that more advanced imaging is of great interest, this will require design of a new construct, establishment of a new stably expressing cell line followed by thorough characterization of the phenotype and transport properties. As such, we hope the Editor and reviewer agree that this is outside the scope of the current manuscript.

We have repeated the imaging study using the same timeline as in the HERA protocol, and a lysotracker was included. In addition to a more thorough determination of the co-localization of IgG variants and FcRn positive endosomes and lysosomes, we have used Imaris spot co-localization analysis. Figures (Figure 3a-h) and Results (page 6) have been updated.

5. In figure 4, microscale thermophoresis is used to determine the affinity between FcRn and different antibodies. Would a cellular based assay where FcRn is in its natural conformation and is present together with B2m be a better method of analyzing affinity?

This is an interesting question. As FcRn is predominantly located within intracellular endosomes where it engages the ligands as a function of a pH gradient, a comparison of direct binding to cell-surface expressed receptor would be challenging, despite that a minor fraction of the receptor may be transiently located at the cell surface. An alternative would be to express the receptor with a GPI-anchor added to the C-terminal end of the extracellular part of the heavy chain, but then the pH of the medium has to be changed to acidic as to allow binding. Although this would be interesting to explore, we find establishment of such an assay behind the scope of the current manuscript.

6. What is the reason the studied pH now significantly differs in comparison to the previous experiments (5.5 instead of 6)? Would incubating at pH 5.5 make the assay more sensitive?

We chose pH 5.5 as the FcRn-ligand complexes are believed to reach this pH in the endosomal pathway. The binding strength may vary slightly between pH 6.0 and pH 5.5, but the same trend is expected. We used MST to determine the KD of the interactions at pH 5.5 and pH 7.4. However, as binding occurs through a pH gradient within endosomal compartments of the cells, we also performed analytical human FcRn chromatography to obtain insights into the dynamics of pH dependent binding to the receptor.

7. An uptake period of 4 hours is chosen, in which recycling already will take place. It would add to the paper to show more details of the kinetics of uptake and recycling.

In the established protocol for HERA, FcRn ligands are added and incubated with the cells for 4 h before the medium is removed by washing. During this period, ligands enter the cells and FcRn binding and recycling or degradation is initiated. We then lyse cell samples to measure the amount of ligand found intracellularly, which is defined as “uptake”. We agree with the reviewer that recycling will already take place. To make HERA a robust and sensitive system however, “accumulation” of a certain amount of the WT antibody inside the cells is needed. The time chosen for the first incubation step (“uptake”) was 4 hours based on an initial experiment where 400 nM IgG1-WT was given to the cells and incubated for 1 h, 2 h, 3 h or 4 h. The highest amount of “uptake” was measured at 4 hours. The data are included in Results and Supplementary Figure 2a-b.

Furthermore, we have addressed to which extent FcRn is involved during uptake. This was done by performing HERA on cells treated with control siRNA or siRNA targeting the FcRn HC that were given either WT hIgG1 or the YTE/KF mutant. The cells were incubated with the antibodies for 15 min, 30 min, 1 h, 2 h, 3 h or 4 h. Cells were then lysed and the amounts present inside the cells quantified by ELISA. The results show that uptake of WT was not affected by downregulation of FcRn, while YTE/KE was taken up less well. Thus, these findings show that uptake of antibodies with no affinity for FcRn at neutral pH occurs in an FcRn-independent manner, while antibodies with FcRn binding at neutral pH engage the receptor at or near the cell surface followed by internalization. The data have been added to Results (page 8), Supplementary Fig. 6a-b and Discussion (page 12).

8. In figure 8, a saturating amount of antibody is added to show that albumin is still able to be properly recycled by FcRn. It would add to the quality of the paper to show that in a context of saturation with HSA, recycling of antibodies is also taking place.

We thank the referee for this question. We have performed HERA experiments where WT anti-NIP hIgG1 was given alone or in the presence of WT HSA or an engineered HSA variant with four mutations that make it bind more strongly and in a less pH dependent manner to FcRn. We found that the presence of the engineered HSA variant did not affect uptake of hIgG1, while in the presence of WT HSA, uptake was reduced by roughly 50%. Despite this, the amounts taken up of hIgG1 was as efficiently recycled as in the absence of WT HSA, while blocking of the albumin binding site actually increased recycling of IgG1. The data have been added to Results (page 10), Fig. 8, and Discussion (page 13).

We have added an ELISA as Supplementary Fig. 8 to show how the engineered variant binds pH independently to human FcRn and compared with WT HSA.

9. Can the authors give explanation why in figure 3 another timeframe is chosen in comparison to the other assays for uptake and recycling?

This concern was also raised by reviewer #2 (4). We have repeated the imaging study with the same timeline as for HERA, and the conclusions remain the same. We did also include a lysosomal tracker to visualize the antibody fractions located to lysosomes. In addition, we have quantified co-localization of the antibodies with the FcRn positive endosomes and lysosomes. The imaging figures (Figure 3a-h and Supplementary Fig. 4a-d) have been updated and Results (page 6) updated in accordance with the findings.

10. Does the HERA score of HSA and K573P HSA correlate also with the half-life? This would be worth adding to figure 7.

We agree with the reviewer that it would be of interest to address the correlation between HERA data and *in vivo* half-life studies for albumin variants. As we only tested 2 variants with FcRn binding properties in the study we did not draw any conclusion on correlation. We have made a large panel of engineered albumin variants (not yet published) that we have screened for their ability to be recycled in HERA. However, in regard to half-life measurements, a major challenge exists when it comes to evaluation of human albumin variants in mice. We have previously shown that mouse FcRn binds very weakly to human albumin, with a binding affinity 30-fold weaker than that of mouse albumin. Thus, FcRn in normal mice prefers to rescue endogenous albumin over injected human albumin. We have also demonstrated that human FcRn binds mouse albumin 5-fold more strongly than human albumin. Clearly, this compromises *in vivo* half-life evaluation of human albumin also in mice Tg for human FcRn. Thus, no reliable rodent model has so far been available for evaluation of human albumin therapeutics. However, recently a new strain that is Tg for human FcRn and where the gene for mouse albumin is deleted (Tg32-Alb(-/-) mFcRn(-/-) hFcRn(Tg/Tg)) has been made by the Roopenian lab, with whom we collaborate. Protocols for half-life evaluation in the new model have been established in our lab, and unpublished results using this strain demonstrate a key role of FcRn. We are now using this new model to measure the half-life of human albumin and engineered variants in the absence and presence of competing albumin. However, these experiments are far from finished. Thus, we cannot yet reveal how HERA on engineered albumin variants correlates with *in vivo* data in human FcRn Tg strains.

11. In the text there is being referred to figure 5C: “to the relative amounts of antibodies remaining inside the cells after additional 4h incubation at pH 7”. It would seem that this should be figure 5F.

We thank the reviewer for this comment. This is correct and we have changed 5C to Figure 5F.

12. Figure legend of figure 1d: What do the researchers mean with “before 0 h”? This seems to be an error and should be corrected.

We appreciated this comment and have deleted “before 0 h sample is collected” from the sentence.

13. Figure 2: Clarity would improve when “concentration antibody” would be added to the x-axis of figure a and b. Also add that figure D-I are at pH 7.4.

We fully agree. “WT hIgG1 added to cells (nM)” has been added to Figure 2A and b. Also, we have clarified that the experiments shown in Fig. 2d-i were done at pH 7.4.

14. Figure 3: It is difficult to understand that the added picture legends indicate green or red signals signals for FcRn/Antibody, but are displayed in white, but that these colors are shown in the third panel. Furthermore, in the middle panel of figure 3d an insert is still visible without any function.

We thank the referee for this comment. To clarify this, the imaging pictures for each channel separately are now shown in grayscale for readers to more easily see the cellular distribution. Especially red and blue colours are not clear on a black background. The text has been changed to FcRnEGFP/Lysotracker/hIgG in the appropriate colours.

15. Figure 4: Why is the analytical hFcRn affinity chromatography profile called a “SEC profile”? It seems that this is an error and should be corrected.

Correct. This is a typo. We have changed “SEC profiles” to “elution profiles”.

16. Figure 5: Why are there a lot less data points in figure D-F if these points are derived from A-C? In addition, a part of the bar in figure 5F is missing. Finally, the data in figure 5b and figure 5g seem to show groups for the same antibody behaving very differently. Please clarify.

We thank the referee for this comment. To calculate the relative difference between the IgG variants (Fig. 5d-e), the average values of uptake, recycling and residual amounts from one independent experiment was used. Therefore, less data points are found in Fig. 5d-f, whereas Fig. 5a-c show results derived from 4 independent experiments performed in triplicates. The bar of Fig. 5f has been divided into lower bar (0-20, 60% of Y axis) and top bar (20-80, 40% of Y axis) for better visualization of the results.

Results shown in Fig. 5a-c were obtained by performing HERA at neutral pH, while results shown in Fig. 5 g-i. were obtained by performing HERA at acidic pH. When the medium is acidified, the assay can be sensitized, but then each antibody can engage the receptor at the cell surface and be internalized and accumulate as the antibodies will not be released efficiently into the medium during incubation. Also, when uptake is performed under acidic conditions, the influence of distinct binding properties detected for the variants at neutral pH is less well pronounced. Thus, differences are observed between the variants in Fig. 5a-c and 5g-I, which reflect that the first set of experiments were done with an initial incubation step at pH 7.4 while the other panel was run at 6.0.

17. Figure legend figure 6: I think before you mention “5 animals”, some text was deleted. This should be corrected.

The sentence has been changed to “The antibodies were administrated as a single i.v. injection to 5 mice per group.”

18. Figure 5: Why is statistical analysis only applied for pH 7.4?

Thank you for this notion. We have added statistical analysis to the experiments done at pH 6.0 as well.

19. Figure 8: Please draw a line for significance between WT and YTE/KF.

We have added a line between WT and YTE/KF.

20. Although the text is clearly written most of the time, a number of spelling/grammatical errors appear in the text.

We have made an effort to clean up the text.

Reviewer #4:

1. *The in vitro cell-based studies show that IgG uptake depends on binding affinity of IgG-FcRn, as IgG with increased affinity results in increased uptake at both pH 7.4 and 6 (Figure 5). Also, an IgG mutant that doesn't bind to FcRn cannot be uptaken in this cell model (Figure 3). These data highlight an ongoing dispute in the field (whether FcRn is located in the cell membrane or not) and thus demands further experiments before accepting this manuscript.*

3. *It seems that the IgG mutant that does not bind to FcRn (IgG IHH) are not uptaken by HMEC1-hFcRn cells (Figure 3). This observation is mentioned, somewhat discussed (page 6, line 152-153; and page 11, line 290-291) but not resolved. In parallel, IgG mutants that have higher affinity to FcRn have higher uptake. These data can be explained only if FcRn is involved in IgG uptake (and indicates that IgG uptake is not simply based on non-specific pinocytosis). Also, reviewer disagrees with the argument that the non-visible signal of the uptake of this mutant might be the result of the fast degradation of IgG, as Alexa647 does not fade in the lysosome (as this stain is pH insensitive) and thus could have been visible even if IgG is degraded. If there is no sign of Alexa647 in these cells that means no uptake of Alexa647-labeled IgG. Reviewer requests integrating additional experiments to clarify this important question. Perhaps shorter uptake (eg. 15 min, 30 min, 1 hr, 2 hr at both pH) with appropriate controls should be applied, or using FcRn downregulated cells.*

We fully agree with the reviewer that further in-depth imaging studies are needed to fully understand how FcRn transport its ligands. In this regard, Sally Ward and her colleagues have contributed during the last 10-15 years with elegant and advanced imaging studies on how the receptor transports IgG variants with different binding properties. In our case, we will in the future focus on HERA, combined with imaging and other techniques, to understand how FcRn recycles and transports its two ligands and therapeutics built on either molecule. In the current manuscript, we included some imaging to pin-point the effect of altering the pH of the medium, so as to force accumulation/uptake in the cells, as this has been done by others. However, the main focus of the current manuscript is the establishment of HERA as a sensitive and robust system to screen non-labeled IgG and albumin variants combined with a simple ELISA readout. In addition, we highlight that HERA can be combined with imaging studies as the over-expressed receptor contains EGFP, and we also discuss the dispute in the field on whether FcRn is expressed on the cell surface or not, and if, how that will effect uptake. We agree that this is of particular interest and should be addressed, however, using more advanced imaging techniques such as TIRF. However, we have included one additional experiment, that we believe has strengthened our manuscript and addresses the question of whether FcRn contributes to uptake. This was done by performing HERA on cells treated with control siRNA or siRNA targeting the FcRn HC that were given either WT hIgG1 or the YTE/KF mutant. The cells were incubated with the antibodies for 15 min, 30 min, 1 h, 2 h, 3 h or 4 h. Cells were then lysed and the amounts of antibody present inside the cells quantified by ELISA. The result showed that uptake of WT was not affected by down regulation of FcRn, while YTE/KF was taken up less well. Thus, these findings show that uptake of antibodies with no affinity for FcRn at neutral pH occurs in an FcRn-independent manner, while antibodies with FcRn binding at neutral pH may then engage the receptor at the cell

surface followed by internalization. The data have been added to Results (page 8) and Supplementary Fig. 6a-b.

Furthermore, the imaging experiments were repeated, using the same time points and concentration of IgG as in HERA. In addition, co-localization between hIgG1 and FcRn positive endosomes or lysosomes were quantified using Imaris spot co-localization analysis (Fig. 3a-h; supplementary Fig 4a-d). The new results did not change the previous conclusions. Figure 3 has been replaced and the result updated.

The data show that roughly 3 times more endosomes are positive for WT hIgG1 than the IHH variant when uptake was performed for 4 hours at pH 6. Under this condition, WT hIgG1 can engage FcRn at or near the cell membrane, and thus the receptor will be actively involved in uptake. This will lead to more efficient uptake/accumulation of IgG WT at pH 6 than at pH 7.4. In contrast, IHH will not be actively internalized via FcRn at pH 6, since it does not bind the receptor. Thus, uptake is markedly different between the antibody variants at pH 6 (Fig. 3a-d). When the uptake phase was done at pH 7.4, WT hIgG1 will be taken up in an FcRn-independent manner (as confirmed by the siRNA experiments described above). As such, at pH 7.4, both WT IgG and IHH was taken up by pinocytosis, and the amounts of uptake were more similar between the two variants (Supplementary Fig. 4a-d) compared to the experiments done at pH 6 (Fig. 3a-h). We fully agree that hIgG1-IHH will be visible inside cells due to the pH insensitive nature of Alexa647. We therefore added a lysosomal marker to visualize sorting of the antibodies in lysosomes, which revealed that the IHH variant shows higher degree of co-localization with lysosomes compared with the WT.

4. Figure 2. legend should indicate the function of the different monoclonal antibodies (DVN24, ADM31, ADM12) – although manuscript provides necessary information in this regards, this info in figure legend would help readers to understand this figure without reading the related text.

We thank the reviewer for this comment and have added information on the monoclonal antibodies used.

5. Figure 5. scales are different and thus confusing a bit (especially inlet a versus g, or c versus i); also, reviewer would like to see data of uptake of IgG-IHH (that does not bind to FcRn).

The scales for the Figures have now been corrected. Uptake of hIgG1-IHH was quantified by imaging at pH 6.0 and 7.4 (Figure 3 and Supplementary Figure 4).

6. Figure 8. it seems that descriptions of inlet “a” and “b” are switched: reviewer thinks that “a” indicates relative uptake while “b” refers to relative recycling” (as in the Y axis of the referred figures)

We have changed the text in accordance with this.

7. Table 2 presents serum half-life data in wt and humanized FcRn mice. Please explain why IgG mutants have been analyzed in two different mice, i.e. Tg32 hemizygous (only one allele express human FcRn) and Tg32 homozygous (both alleles express human FcRn). Theoretically, IgG half-life depends on many factor, including the level of FcRn expression.

WT and engineered IgG1 variants were evaluated in Tg32 hemizygous mice as this is the preferred state-of-the-art pre-clinical model to be used for comparison of WT and engineered human IgG variants. We chose this mouse model following advice from JAX Service and Professor Derry Roopenian. Using this mouse strain, differences in FcRn binding kinetics translate into differences in half-life values that are more distinct than what can be obtained using Tg32 homozygous mice, especially, under conditions where there is no competition from endogenous IgG.

The only exception in our study is the YTE/KF (“Abdeg”) variant that was evaluated in homozygous Tg32 mice. The reason for this is that the variant was initially included in a related study where it was compared with other variants that are not included in this manuscript. At that time, this strain was the only one that we had at hand. As there is no strong evidence that the Abdeg variant should behave very differently in the two strains as it binds in a pH independent manner, we were advised to not include this in the panel tested in the hemizygous mice.

REVIEWERS' COMMENTS:

Reviewer #3 (Remarks to the Author):

The revised manuscript of Grevis et al was significant improved upon revision. Most of my concerns were correctly addressed. Also figure 3 is much better now, although there are some small mix ups: the results section mentions 50 - 60 % confluency (line 145-146), but the method section mentions 70 % confluency. And in the text it is stated that cells are incubated for 4 hrs at pH 6, pictures are taken, and after an additional 4 hrs at 7.4 again pictures are taken, while the figures says 0 and 4 hrs. Please adjust to 4 hrs and 8 hrs.

When these small errors are corrected, the manuscript is acceptable for publication in my opinion.

Reviewer #4 (Remarks to the Author):

The authors have made several novel experiments and meaningfully improved the manuscript. I therefore recommend publication this manuscript in Nature Communications.

Reviewer #3: *The revised manuscript of Grevys et al was significant improved upon revision. Most of my concerns were correctly addressed. Also figure 3 is much better now, although there are some small mix ups: the results section mentions 50 - 60 % confluency (line 145-146), but the method section mentions 70 % confluency. And in the text it is stated that cells are incubated for 4 hrs at pH 6, pictures are taken, and after an additional 4 hrs at 7.4 again pictures are taken, while the figures says 0 and 4 hrs. Please adjust to 4 hrs and 8 hrs. When these small errors are corrected, the manuscript is acceptable for publication in my*

opinion.

We appreciated this comment and have corrected the percentages of confluency in both the results and methods sections. In addition, we have adjusted the time points to 4 and 8 hours.

Yours sincerely,

Jan Terje Andersen & Inger Sandlie